# Numerical Investigation of Different Core Topologies in Sandwich-Structured Composites Subjected to Air-Blast Impact

Marcel Walkowiak [ID], Ulf Reinicke and Denis Anders *[ID]

Group for Computational Mechanics and Fluid Dynamics, Cologne University of Applied Sciences (TH Köln), 51643 Gummersbach, Germany
* Correspondence: denis.anders@th-koeln.de; Tel.: +49-(22)-6181966372

**Abstract:** Air-blast loading is a serious threat to military and civil vehicles, buildings, containers, and cargo. Applications of sandwich-structured composites have attracted increasing interest in modern lightweight design and in the construction of dynamic loading regimes due to their high resistance against blast and ballistic impacts. The functional properties of such composites are determined by the interplay of their face sheet material and the employed core topology. The core topology is the most important parameter affecting the structural behavior of sandwich composites. Therefore, this contribution presents a thorough numerical investigation of different core topologies in sandwich-structured composites subjected to blast loading. Special emphasis is put on prismatic and lattice core topologies displaying auxetic and classical non-auxetic deformation characteristics in order to illustrate the beneficial properties of auxetic core topologies. Their dynamic responses, elastic and plastic deformations, failure mechanisms, and energy absorption capabilities are numerically analyzed and compared. The numerical studies are performed by means of the commercial finite element code ABAQUS/Explicit, including a model for structural failure.

**Keywords:** lightweight structures; sandwich-structured composites; auxetic; blast impact; energy absorption



## 1. Introduction

Increasing requirements concerning environmental impact, product quality, and security lead to new challenges for lightweight design [1–4]. In particular, the thrust towards renewable energy and the expansion of electro-mobility encourage novel trends in the development of materials and multi-functional structures. In this context, the stability of material structures (e.g., body frames [5] and battery-pack enclosures [6]) subjected to dynamic loading regimes attracts increasing interest. For this reason, the present contribution puts a special focus on the blast resistance of lightweight structures. Numerous studies have demonstrated the improved performance of structural components based on sandwich-structured composites in the case of blast loading due to detonation processes (in water or air) compared to monolithic panels of the same weight [7–10]. This superiority is expressed both in the lower level of deflection with respect to the rear face and in the reduction in the forces transmitted to the substructure. The main reasons for this enhanced blast resistance are the increase in energy dissipation due to plastic deformations of the core and the top surfaces, as well as—mainly in the case of underwater explosions—a reduction in the momentum introduced into the structure due to fluid–structure interactions.

The load-carrying behavior of sandwich-structured composites under the impact of explosions exhibits fundamental differences compared to quasi-static loading. According to [11], dynamic effects, such as the rate-dependent material behavior, the inertia stabilization of the core walls/bars, and the mass inertia resistance, provide for an increase in stiffness in the core area. The inertia-stabilizing effect on the core elements leads to an increase in the dynamic buckling resistance; this effect becomes more significant with increasing impact loading.

The core topology also determines wave propagation through the structure and plastic distortions, as well as the type and shape of the plastic buckling processes. To our knowledge, a thorough classification of the dynamic performance of auxetic core geometries compared to conventional approaches does not yet exist; this gap in the literature serves as a motivation for this contribution. A novel aspect of this work is the fact that both open and closed prismatic cores, as well as lattice cores, each in different auxetic and non-auxetic designs, were investigated in this context. Please note that this contribution is essentially based on the preliminary work of the first author [12].

## 2. State of the Art

Due to the high level of numerical effort involved, some authors refrain from an exact modeling of resolved core structures and describe their behaviors with the help of homogenization techniques [13–15]. Such continuum mechanical approaches inevitably lead to simplified representations of the relevant behaviors and failure mechanisms, with an associated loss of accuracy. For example, these approaches cannot capture local (buckling) phenomena over the core height or their influence on overall structural behavior. In addition, the homogenized input parameters of any core variants must be determined and calibrated at great expense, either by means of experimental data or detailed mapped and meshed unit cells.

In addition, simplified analytical approaches exist to describe core geometry and core properties. Deshpande and Fleck [16–18], for example, used analytical models for the description of the structural response of sandwich beams under blast or impact loading. They investigated different core topologies (honeycomb, pyramidal, prismatic diamond, and metal foam) and characterized the different variants on the basis of their transverse compressive strength (out-of-plane) and their tensile strength in the longitudinal direction as a function of the relative density. Comparisons with finite element calculations show good agreement. However, the analytical models often underestimate (especially at high core strengths) the maximum deformations of the backside and provide overly high values for the reaction forces in the supporting substructure [19]. The focus on a beam-like system additionally limits the general validity.

Due to increased computer capacities, complete topology models of the discrete sandwich structures were considered in recent studies [20–22]. These studies focused primarily on the investigation of various core topologies, the recording of fluid–structure interactions in a fully coupled analysis, and the description of the failure mechanisms of the core and top surface areas as well as the connecting regions.

Comparative studies by Dharmasena et al. [23,24] illustrated the superior properties of several core topologies (honeycomb, prismatic, and pyramidal truss cores) in sandwich structures over monolithic panels when subjected to impulsive loading in water. With a relative density of about five percent, the best variants reduced the forces transmitted to the substructure by 25% compared to a monolithic panel of the same weight. Moreover, pyramid-shaped lattice core structures embedded in a multi-layer design achieved up to 30% lower values [24]. In addition, the adjustment of the relative core density allowed selective control of the transferred pressure [25,26], whereby lower densities tended to mean lower amounts. Homogenization approaches and simplified load-modeling strategies (uniform initial velocity of the outer face) show acceptable agreement with the results of comparative experiments carried out under planar pressure blast loading after calibration has taken place [23,24].

Chen et al. provided a comprehensive study of auxetic double arrowhead honeycomb core sandwich panels for performance improvement under air-blast loading [27]. Their numerical results reproduced former experimental investigations [28] and confirmed the material concentration effect of auxetic double arrowhead honeycomb cores induced by a negative Poisson's ratio. According to the simulation results, the best design strategy for improving the blast performance of this kind of auxetic core structure depends upon core relative densities. The adoption of more highly inclined angles could reduce the level of deflection for the panel center more efficiently at a low relative density. For higher levels

of relative core density, it is more efficient to decrease the horizontal distance instead of increasing the incline of the angles. With regard to the total energy dissipation, panels with thinner core webs are able to dissipate more plastic energy at low relative density, whereas those with narrower incline angles are superior at high relative density.

However, most studies only focus on one class of core geometries. As a result, a systematic classification and comprehensive comparison of core topologies for sandwich structures are lacking. The present contribution aims to close this gap.

## 3. Classification of Core Topologies

As already explained in the previous sections, the core of a sandwich construction contributes significantly to the functional performance of the overall structure. It determines quasi-static load-bearing capacity due to its shear stiffness and buckling resistance in the vertical direction. In addition, the core topology affects system behavior in dynamic ballistic and/or blast loading scenarios. Usually, there are three fundamentally different strategies for core designs:

1  Polymeric, ceramic, and metal foams with stochastic void distribution (open-cell foams or closed-cell foams);
2  Periodic prismatic core geometries (open or closed);
3  Periodic lattice-type core geometries (lattice cores).

Figure 1 shows an example of two elementary prismatic core structures with auxetic base geometry. The design on the right belongs to the family of so-called honeycomb cores (closed-cell cores). Here, the longitudinal direction of the cells always coincides with the thickness direction of the core, and a closed, periodic interior space is created in the plane. The alternative design on the left-hand side is open on one side and allows, for example, oriented mass transport in one of the surface directions (unidirectional open-cell cores). In this case, the longitudinal direction of the cells is parallel to the two cover layers.

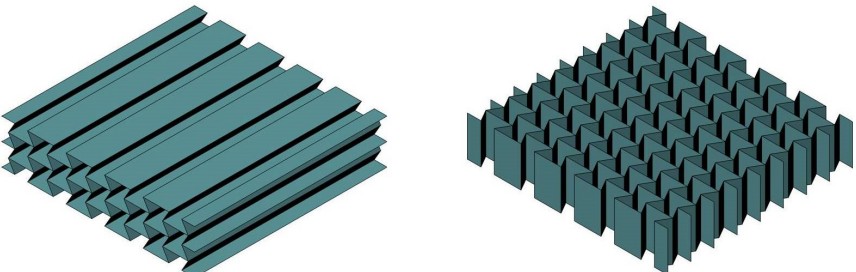

**Figure 1.** Examples of prismatic core structures based on the standard auxetic topology. (**Left**) open-cell core. (**Right**) closed-cell core.

A periodic core design that is open on all sides can only be realized by means of lattice-type cell elements (so-called lattice cores). Three-dimensional lattice structures provide versatile options for parameter-based topology optimization. Figure 2 illustrates a lattice core design based on a three-dimensional auxetic unit cell.

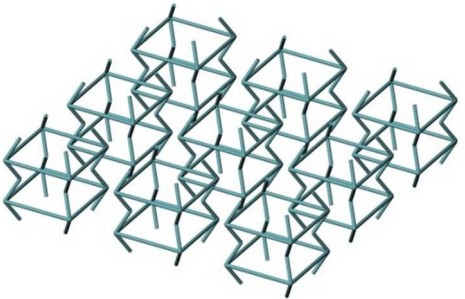

**Figure 2.** Example of a lattice-type core structure based on a three-dimensional auxetic unit cell.

The mechanical deformation behaviors and, thus, the mechanical properties of the three elementary core variants differ in fundamental ways. The manufacturing process for conventional polymeric, ceramic, and metal foam usually generates a randomly distributed void structure with irregular and corrugated cell walls or cell webs.

During loading, these cell components are primarily subjected to bending stresses [29]. In addition, closed cells usually have very low relative membrane load-bearing capabilities and fail due to the rapid onset of buckling or cracking effects [30]. Imperfections, defects, and inhomogeneities caused by the manufacturing process lead to the comparatively low mechanical strength and stiffness of foam cores [31]. Many analyses of this foam's properties, the parameters that can be used to adjust its mechanical properties, and its use in sandwich construction exist in the literature. Therefore, foam cores are not the subject of this study.

However, core structures with conventional periodic prismatic structures are primarily subject to a completely different deformation mechanism over wide load ranges. The regular cells provide a spatial folding structure with a dominant vertical axial orientation and components with a high relative membrane load-bearing capacity. Their mechanical properties depend primarily on the material and the buckling strength of the cell walls. In principle, this load-bearing behavior leads to higher stiffness and strength values than what is observed in foam cells that are mainly subject to bending stress.

Similar correlations can also be deduced for the other core variants. However, the specific choice of geometry parameters for unilaterally open prismatic core cells, as well as for lattice cells, strongly affects their membrane and bending load-bearing capacities, more so than with other core types. So, it is not possible to identify a dominant load-bearing behavior in advance.

### 3.1. Core Topologies and Their Parameterisation

New practicable, precise, and cost-effective manufacturing techniques [32–35] led to a steadily increasing interest in innovative and efficient core geometries in recent years. This resulted in a large number of topology studies investigating the influence of geometric parameters on mechanical properties under various physical conditions and loading scenarios. However, hardly any attention is paid to macroscopic auxetic structuring. The focus here is, therefore, on classifying the performance of auxetic topologies in the broad performance spectrum of conventional non-auxetic core variants.

The most common closed prismatic core topologies include square honeycomb cores and honeycomb cores composed of regular hexagonal cells. The behavior of these cores as a function of their relative density, the ratio of core height to cell size, the number of cells, and the impact of the use of different materials have been documented for a variety of impacts. For a vertical quasi-static compressive load, there is a linear relationship between the relative density, the maximum density, and the maximum compressive strength of the core construction. The influence of the cell size ratio is comparatively small [36]. The same applies to shear loading in the plane; the shear strength increases linearly with the relative density, while the cell wall dimensions play a subordinate role in relation to the core thickness [37]. The use of composite materials leads to qualitatively similar results. Here, the fiber direction of the carbon-fiber-reinforced polymer (CFRP) laminate decisively influences the system response [38–40]. With the help of a hierarchical core structure, which includes a sandwich construction for the cell walls, the buckling strength and, thus, the compressive strength can be significantly increased compared to conventional composite cores [41]. Dynamic compression tests with changing load application angles on aluminum honeycomb cores also proved that an increase in dynamic compressive strength and an almost invariable increase in shear strength take place with increasing impact velocity [42].

Figure 3 shows sections of the honeycomb cores examined in this contribution, including the associated unit cells and their respective topology parameters. In addition to the cell width $b_{uc}$ and the cell height $h_{uc}$, the inclination angle $\alpha$ and the sheet thickness $t_c$ of the cell walls also determine the pattern geometry and mechanical behavior. The depth

of the vertical prismatic cells results from the thickness $h_c$ of the core layer. In addition to the conventional honeycomb core, an isogrid variant (based on the regular honeycomb pattern), a quadratic core, and the standard auxetic honeycomb are considered.

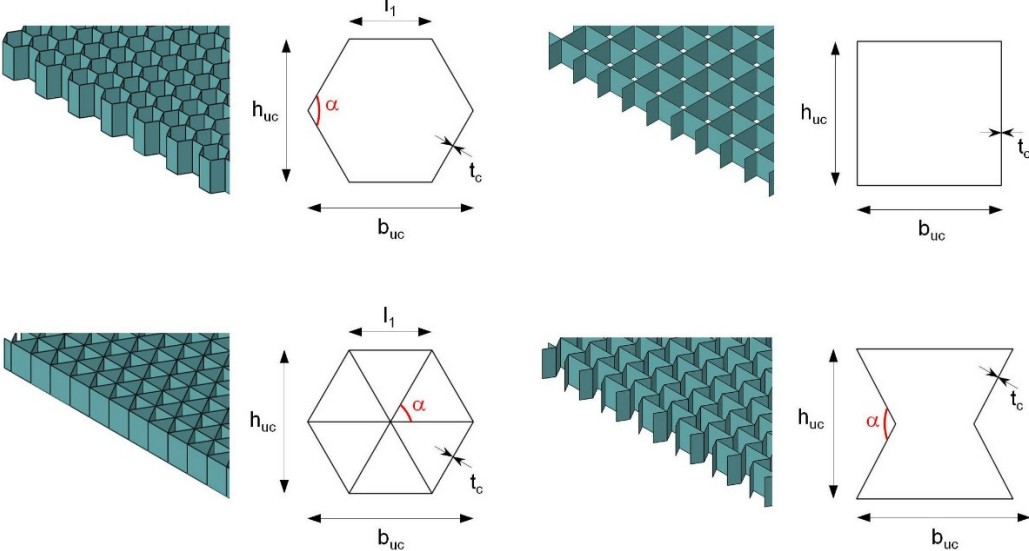

**Figure 3.** Sections of prismatic closed-cell honeycomb cores with their respective unit cells.

Open prismatic cores form a related class of core topology. The extrusion direction of the underlying unit cells is located parallel to the cover surfaces, resulting in core designs that are open on one side. The core geometries shown in Figure 4 are widely used in such cases; in principle, the same geometry parameters used for their closed counterparts can be used to adjust their mechanical properties. Extensive studies have already been conducted on this core group to determine their mechanical behavior in different loading situations.

For example, triangular and diamond-shaped core designs have lower compressive and transverse shear stiffness than conventional honeycomb designs, regardless of the relative density. Only shear loads in the longitudinal direction can be absorbed in comparable orders of magnitude [43]. Structural optimization studies show that the weight efficiency of triangular and diamond-shaped core geometries is comparable to that of conventional honeycomb cores only at high compression load levels. Dependence on the load position is particularly pronounced for the triangular variant. In its case, only a bending load orthogonal to the longitudinal axis of the cell leads to acceptable results due to the increased buckling stiffness. The performance of the rhombic pattern is almost independent of the load orientation with appropriate individual parameter selection [44]. Experiments by Valdevit et al. [45] produced overview maps for the prediction of the relevant failure mechanisms in triangular topologies as a function of geometrical configuration and actions.

As with the closed honeycomb cores, a hierarchical structure also makes it possible to improve the performance of the overall structure. To this end, Kooistra et al. [46] designed core walls as a triangular sandwich construction (second order). While conventional structures with monolithic cell walls (first order) fail due to the buckling processes or plastic flow processes of the core components, six potential failure mechanisms are now taken into account in second-order structuring. Hierarchization results in significantly higher compressive and shear strengths for the same weight (relative core density $\rho_c < 5\%$). Côté et al. [47] further identified the critical regions for the fatigue of diamond cores under transverse shear loads. In their study, depending on the relative density, the immediate core components in the corner regions or the connections between the core and face regions were the limiting factors. The fatigue-related failure of interfaces was observed in the course of this investigation in a number of other core types.

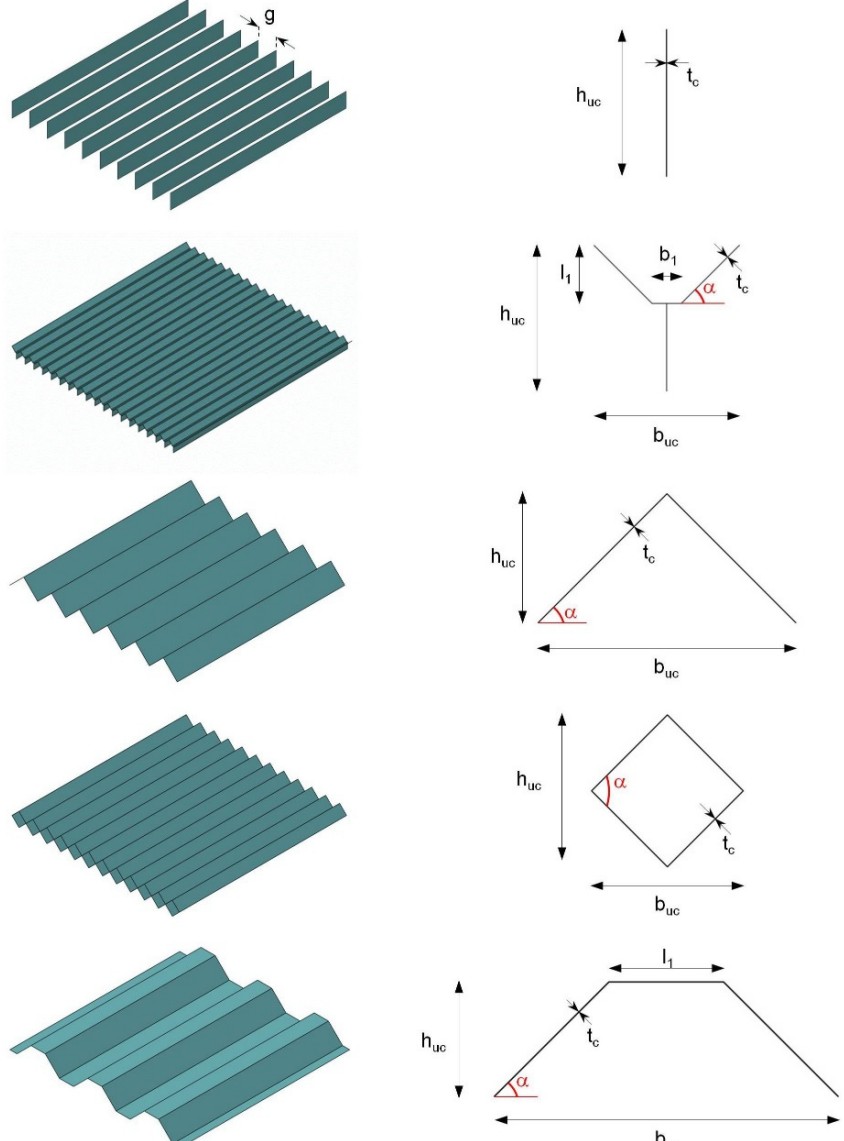

**Figure 4.** Sections of conventional prismatic open-cell cores with their respective unit cells (one layer, 1L). From top to bottom: I, Y, triangle, diamond, and container truss core.

Experimental studies on sandwich specimens by Rubino et al. [48] indicated similar load-displacement behavior for Y-cores and triangular configurations. This was true both for a quasi-static three-point bending load with different boundary conditions and for time-independent plane indentation tests perpendicular to the sandwich faces. FE simulations by Rubino et al. [49] showed, in good agreement with real test results, relatively low compressive and transverse shear strengths for Y-cores compared to their longitudinal shear capacity; this result was due to the plastic bending deformations of the core components. Their energy dissipation capacity under compressive stress was much higher than that of conventional and equal-weight sandwich components with interior fillings of metal foams. Pedersen et al. [50] recognized, for most of the Y-core variants relevant in practice, a constant load plateau with progressive compression (out-of-plane) of the sandwich construction; they confirmed its capacity for energy dissipation.

Figures 5 and 6 present typical topologies from the class of two-layer prismatic and one-side-open sandwich cores, including the underlying unit cells and their geometry parameters.

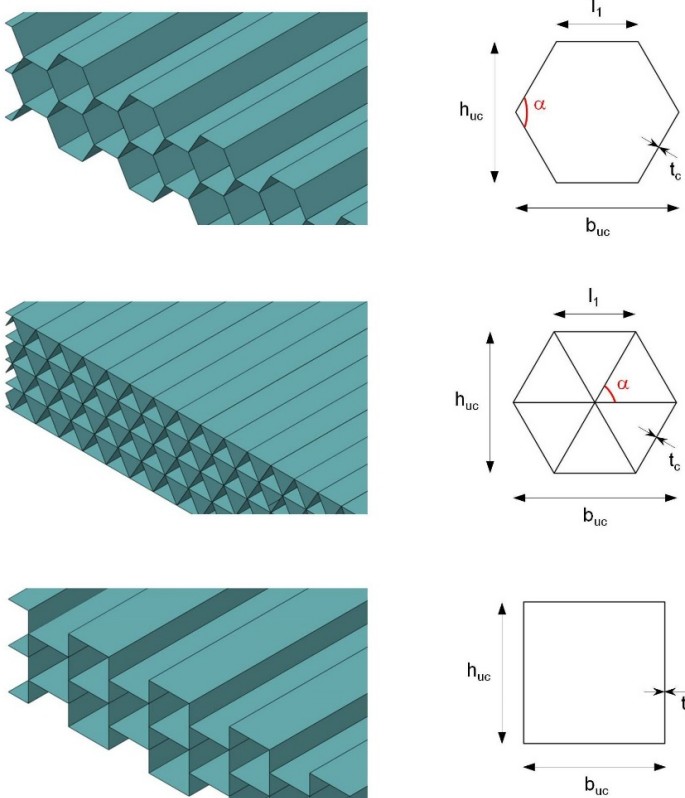

**Figure 5.** Sections of non-auxetic prismatic and one-side-open core structures with their associated unit cells (two-layer, 2L). From top to bottom: honeycomb, isogrid, and quadratic core.

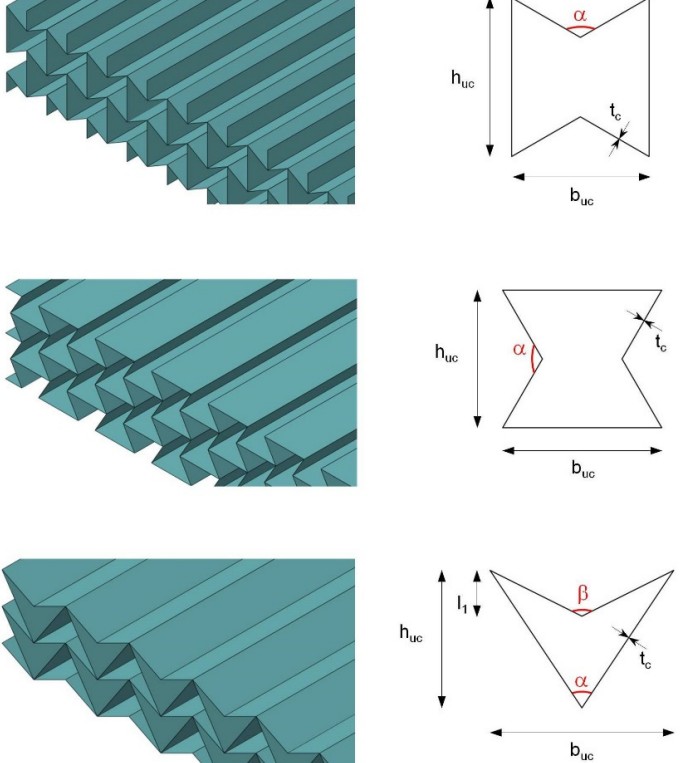

**Figure 6.** Sections of auxetic prismatic and one-side-open core structures with their associated unit cells (two-layer, 2L). From top to bottom: standard auxetic core (90° rotated), standard auxetic core, and dart core (concave deltoid).

A new feature is the use of auxetic variants in Figure 6. Besides the standard auxetic cell in the upright and 90° rotated positions, the behavior of a core with concave deltoid cells (Figure 6, below) is of special interest due to its two incline parameters. Here, the mechanical behavior and the degree of auxeticity can be specifically controlled, within certain limits, via two independent angle settings; this offers a potential advantage over patterns with fewer influencing variables.

The last category of cores consists of the lattice core variants. Like the prismatic versions, these cores have high weight-related strengths with low relative densities, as well as a high specific energy absorption capacity. Furthermore, the fully open core area creates a great deal of additional functionality. Application capabilities, including as a heat exchanger with a cooling medium flowing through it or as a storage structure for electrical energy (in which the core topology serves as one of the electrodes), are described in the literature; in addition, the manufacturing costs are lower than those of conventional honeycomb cores [51]. Overviews of the multifunctional application possibilities and the application-related design requirements of these cores were presented by Evans et al. [52,53].

So far, the lattice topologies shown in Figure 7 are mainly used in practical applications. Not listed are the square and rhombic lattice cores, whose mechanical properties were described by Moongkhamklang et al. [54,55]. For lattice cores, the structural behavior under compressive loading is characterized by elastic buckling processes and under shear loading by tensile failure in the lattice bars. When the bars are designed as silicon carbide–titanium composites, even higher shear strengths can be achieved than with honeycomb cores made of carbon-fiber-reinforced polymer (CFRP). The performance capabilities in compression processes are comparable. However, the increase in load-bearing capacity is accompanied by a loss of ductility and the risk of spontaneous failure.

A comparison of the two lattice geometries finds higher shear strength but lower tensile strength (in-plane) for diamond lattices [56]. Under compressive loading (out-of-plane), both variants react similarly. Sandwich elements subjected to bending stress achieve better performance due to the more favorable shear behavior when using diamond-shaped cells.

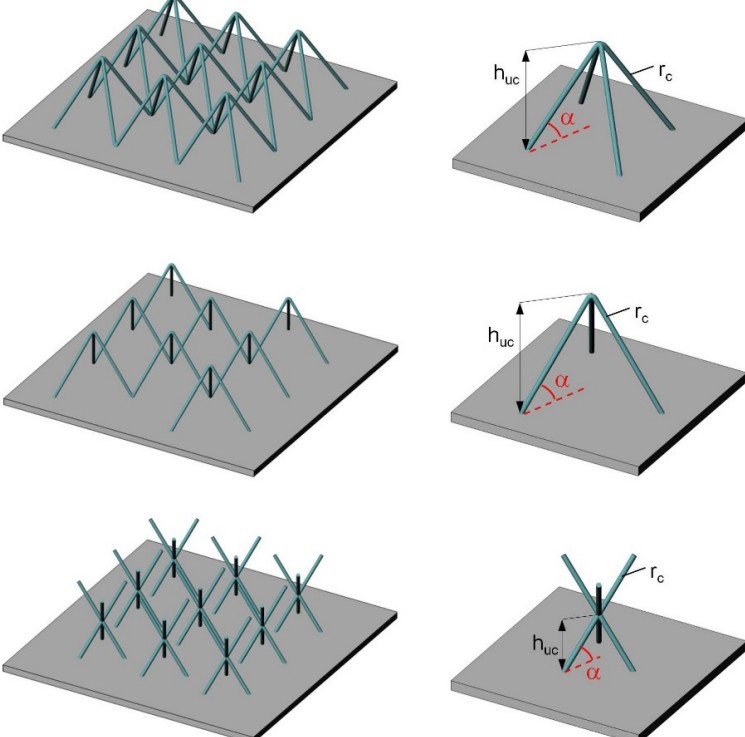

**Figure 7.** Sections of conventional one-layer 3D lattice core variants with their associated unit cells. From top to bottom: pyramidal core, tetrahedral core, and 3D Kagomé core [57].

Studies by Wadley et al. focused on determining the mechanical properties of sandwich constructions with pyramid-shaped lattice core structures. The behavior of these sandwich structures under vertical compressive loading, as well as under longitudinal and transverse shear loading, was documented in detail [58,59]. In addition, the geometry-dependent failure patterns were investigated in [60], taking into account multi-layer core assemblies under in-plane compressive forces. An increase in the mechanical performance of most lattice core structures can be achieved by using circular hollow sections for the lattice bars [61]. Similarly, the use of carbon-fiber-reinforced polymer as a core material promises higher performance potential [62]. Cyclic loading shows that, under shear stress, there is a high susceptibility to fatigue and failure in the brazed or welded connection areas between the core and the cover surface. However, hardly any structural problems were observed during compression processes [63]. Studies on the imperfection sensitivity of the core geometry confirmed the joints as critical regions. Deliberately introduced imperfections in the connection regions led to a loss of shear strength and shear stiffness but did not show any effects in the case of vertical compressive stresses [64].

Since finite-element-based simulations of complex core structures are very computationally expensive in terms of time and CPU capacity, the number of topologies considered in this contribution had to be limited. For this reason, a direct comparison with the previously described variants is not possible for the lattice cores. Rather, the focus here is on a subset of the novel cell geometries shown in Figure 8. The uppermost topology corresponds to the equivalent of a three-dimensional honeycomb cell and is characterized by positive values of the lattice cell angle $\alpha$. With $\alpha = 0°$, cube-shaped cells without inclined cell components are created. A three-dimensional auxetic structure results, with a value of $\alpha$ in the negative value range (Figure 8, center). Here, a compression in one of the three spatial directions also leads to a contraction of the cell in the two remaining directions. Slight modifications transform the conventional pyramidal geometry into a potentially auxetic structure (Figure 8, below). A central compressive load (such as one that occurs in a multi-layer structure) also tends to cause a contraction in the surface directions.

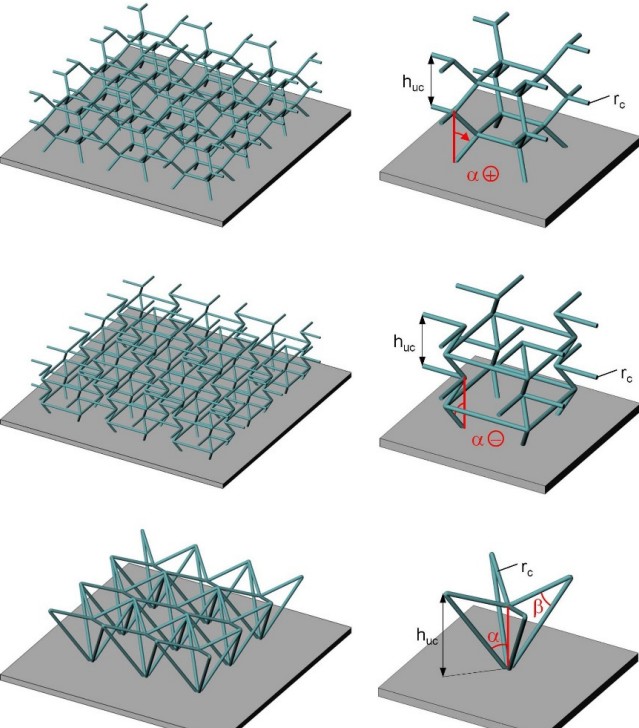

**Figure 8.** Three-dimensional lattice core variants with associated unit cells and geometry parameters (one-layer, 1L). From top to bottom: non-auxetic honeycomb cell, auxetic honeycomb cell, and auxetic arrow geometry.

### 3.2. Considered Core Topologies for Numerical Studies

The primary goal of being able to evaluate and compare numerous core topologies without an increased standardization effort led to the definition of some constant geometry parameters for all sandwich configurations in the run-up to the simulations. In addition to the fixed thickness of the face sheets, equal lateral dimensions and a constant core height were also set for all configurations. Furthermore, a constant relative density was set for all core variants; consequently, identical weights for all constructions were guaranteed. On this basis, an initial efficiency evaluation and assessment of the performance of the auxetic cores compared to conventional variants can be made according to the simulation results.

The data in Tables 1–4 describe the considered core topologies and their geometry parameters. The adjustment of the wall thickness $t_c$ of the prismatic cores is performed automatically and—in order to maintain the weight—depends exclusively on the total cell wall area available in the interstitial space. The parameters of the conventional core geometries are based on common values for sandwich structures of this size.

**Table 1.** Topology and parameters: closed prismatic honeycomb cores. See Figure 3.

| No. | Designation | $h_{uc}$ in mm | $b_{uc}$ in mm | $l_1$ in mm | $\alpha$ in ° | $t_c$ in mm |
|---|---|---|---|---|---|---|
| 01 | honeycomb | 15.00 | 17.32 | 8.66 | 120 | 0.2500 |
| 02 | square-honeycomb | 2000 | 20.00 | - | - | 0.3305 |
| 03 | closed isogrid core | 3000 | 34.64 | 17.32 | 60 | 0.1645 |
| 04 | auxetic honeycomb (h-15) | 15.00 | 17.32 | - | 120 | 0.1895 |
| 05 | auxetic honeycomb (h-20) | 20.00 | 23.09 | - | 120 | 0.2498 |
| 06 | auxetic honeycomb (h-25) | 25.00 | 28.87 | - | 120 | 0.3200 |
| 07 | auxetic honeycomb (h-30) | 30.00 | 34.64 | - | 120 | 0.3609 |

**Table 2.** Topology and parameters: one–side-open non-auxetic prismatic cell cores. See Figures 4 and 5.

| No. | Designation | $h_{uc}$ in mm | $b_{uc}$ in mm | $l_1$ in mm | $\alpha$ in ° | $t_c$ in mm |
|---|---|---|---|---|---|---|
| 08 | I-core (g = 20 mm) | 20.00 | - | - | - | 0.6610 |
| 09 | Y-core ($b_1$ = 4 mm) | 20.00 | 20.00 | 8.00 | 45 | 0.3602 |
| 10 | container truss core | 20.00 | 60.00 | 20.00 | 45 | 0.5476 |
| 11 | triangle-core | 20.00 | 40.00 | - | 45 | 0.4674 |
| 12 | diamond-core | 20.00 | 20.00 | - | 90 | 0.2337 |
| 13 | square-honeycomb (2L) | 10.00 | 10.00 | - | - | 0.2203 |
| 14 | square-honeycomb (4L) | 5.00 | 5.00 | - | - | 0.0944 |
| 15 | honeycomb (2L) | 10.00 | 11.55 | 5.77 | 120 | 0.1808 |
| 16 | honeycomb (4L) | 5.00 | 5.77 | 2.89 | 120 | 0.0864 |
| 17 | open isogrid core | 10.00 | 11.55 | 5.77 | 60 | 0.0603 |

**Table 3.** Topology and parameters: one-side-open auxetic prismatic cell cores. See Figure 6.

| No. | Designation | $h_{uc}$ in mm | $b_{uc}$ in mm | $l_1$ in mm | $\alpha$ in ° | $t_c$ in mm |
|---|---|---|---|---|---|---|
| 18 | auxetic (2L) | 10.00 | 11.55 | - | 120 | 0.1421 |
| 19 | auxetic (4L) | 5.00 | 5.77 | - | 120 | 0.0663 |
| 20 | auxetic rotated-90° (4L) | 6.86 | 5.94 | - | 120 | 0.0722 |
| 21 | dart (2L, $\beta$ = 120°) | 12.00 | 13.86 | 4.00 | 60 | 0.0753 |

The data on open lattice core geometries in Table 4 also contains information on the number of unit cells and layers used. From this, the lateral dimensions of the individual spatial cells can be derived. Furthermore, the radius $r_c$ of the lattice bars replaces the wall thickness $t_c$ of the prismatic cores. The design of the cell structure is carried out here exclusively by means of circular solid cross-sections.

**Table 4.** Topology and parameters: auxetic and non-auxetic 3D lattice cores. See Figure 8.

| No. | Designation | Layers | Cell | $h_{uc}$ in mm | $\alpha$ in ° | $r_c$ in mm |
|---|---|---|---|---|---|---|
| 22 | honeycomb_+45°_L1 | 1 | 15 × 15 | 20.00 | +45 | 1.2455 |
| 23 | honeycomb_+45°_L2 | 2 | 30 × 30 | 10.00 | +45 | 0.5897 |
| 24 | honeycomb_+45°_L4 | 4 | 60 × 60 | 5.00 | +45 | 0.2881 |
| 25 | honeycomb_+30°_L1 | 1 | 15 × 15 | 20.00 | +30 | 1.2058 |
| 26 | honeycomb_+30°_L2 | 2 | 30 × 30 | 10.00 | +30 | 0.5736 |
| 27 | honeycomb_+30°_L4 | 4 | 60 × 60 | 5.00 | +30 | 0.2754 |
| 28 | honeycomb_+15°_L1 | 1 | 15 × 15 | 20.00 | +15 | 1.2108 |
| 29 | honeycomb_+15°_L2 | 2 | 30 × 30 | 10.00 | +15 | 0.5507 |
| 30 | honeycomb_+15°_L4 | 4 | 60 × 60 | 5.00 | +15 | 0.2619 |
| 31 | cube_+0°_L1 | 1 | 15 × 15 | 20.00 | 0 | 1.1231 |
| 32 | cube_+0°_L2 | 2 | 30 × 30 | 10.00 | 0 | 0.5131 |
| 33 | cube_+0°_L4 | 4 | 60 × 60 | 5.00 | 0 | 0.2474 |
| 34 | auxetic_−15°_L1 | 1 | 15 × 15 | 20.00 | −15 | 1.0732 |
| 35 | auxetic_−15°_L2 | 2 | 30 × 30 | 10.00 | −15 | 0.4776 |
| 36 | auxetic_−15°_L4 | 4 | 60 × 60 | 5.00 | −15 | 0.2308 |
| 37 | auxetic_−30°_L1 | 1 | 15 × 15 | 20.00 | −30 | 0.9624 |
| 38 | auxetic_−30°_L2 | 2 | 30 × 30 | 10.00 | −30 | 0.4353 |
| 39 | auxetic_−30°_L4 | 4 | 60 × 60 | 5.00 | −30 | 0.2070 |
| 40 | auxetic_−45°_L1 | 1 | 15 × 15 | 20.00 | −45 | 0.8251 |
| 41 | auxetic_−45°_L2 | 2 | 30 × 30 | 10.00 | −45 | 0.3676 |
| 42 | auxetic_−45°_L4 | 4 | 60 × 60 | 5.00 | −45 | 0.1775 |

Since the auxetic and non-auxetic 3D honeycomb unit cells are, in principle, based on the same basic generating model, Table 4 summarizes both variants. As can be seen from Figure 8, negative angles describe the auxetic and positive angles the non-auxetic variants. With $\alpha = 0°$, a conventional cube structure is created. Figure 9 shows side views for single- and multi-layer structures. Figure 10 illustrates the cell shape changes with variations in the grid cell angle $\alpha$.

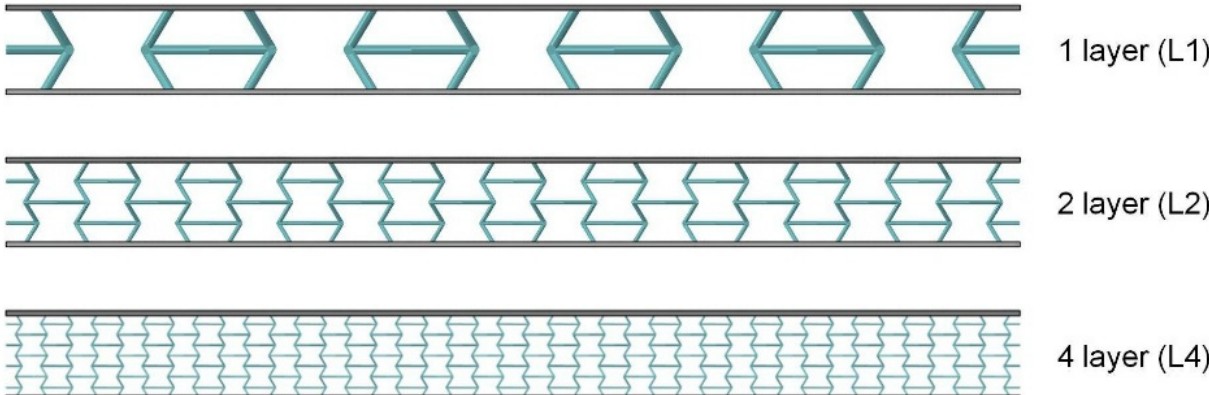

**Figure 9.** Three-dimensional lattice cores in single- and multi-layer configurations (side view in parallel projection).

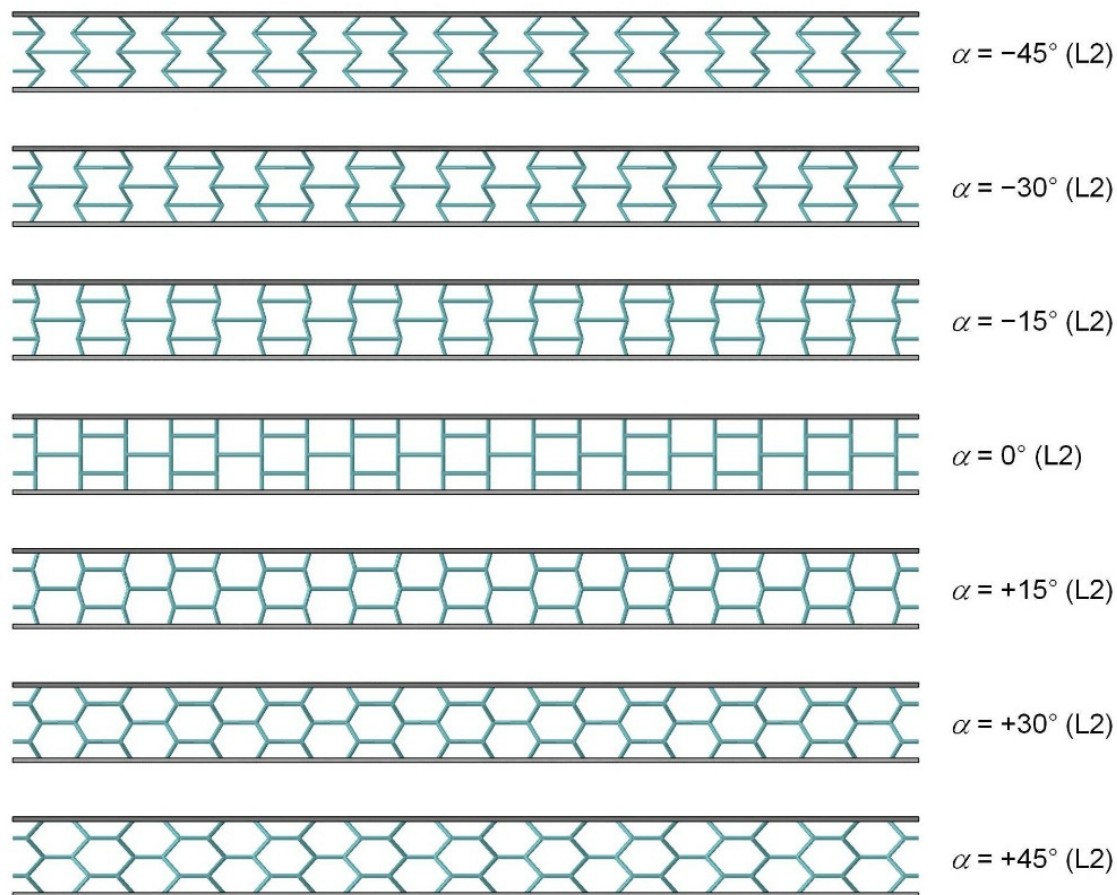

**Figure 10.** Cell shape changes of 3D lattice cores with variations in the grid cell angle $\alpha$ (side view in parallel projection).

## 4. Modeling of the Blast Loading

When energetic materials are ignited, gaseous combustion products are produced that move away from the detonation site at high speed. This rapid expansion causes the creation of a shock wave with spatially and temporally discontinuous pressure, density, temperature, and velocity distributions. The physical states of the media before and after the blast load can be derived from the conservation laws for mass, momentum, and energy and are summarized in the Rankine–Hugoniot jump conditions. The temporal change in the pressure conditions of an ideal shock wave propagating in the free field can be calculated by means of the modified Friedlander equation [65]:

$$p(t) \;=\; p_0 + p_{\max} \cdot \left(1 - \frac{t}{t_+}\right) \cdot e^{\left(-a \cdot \frac{t}{t_+}\right)} \tag{1}$$

At a fixed point in space, the pressure increases abruptly from the ambient pressure $p_0$ to the peak pressure (overpressure) $p_{\max}$ when the pressure front ($t_a$) arrives; then, it decreases exponentially. A positive pressure phase is followed—due to the inertia of the air—by a drop in pressure below the ambient pressure, the so-called suction phase. The duration of the positive phase is designated by $t_+$ and is only a few milliseconds (the main part of the load is usually only tenths of a millisecond). The parameter $a$ describes a dimensionless decay coefficient. Figure 11 shows the characteristic pressure–time curve for wave propagation in the free field.

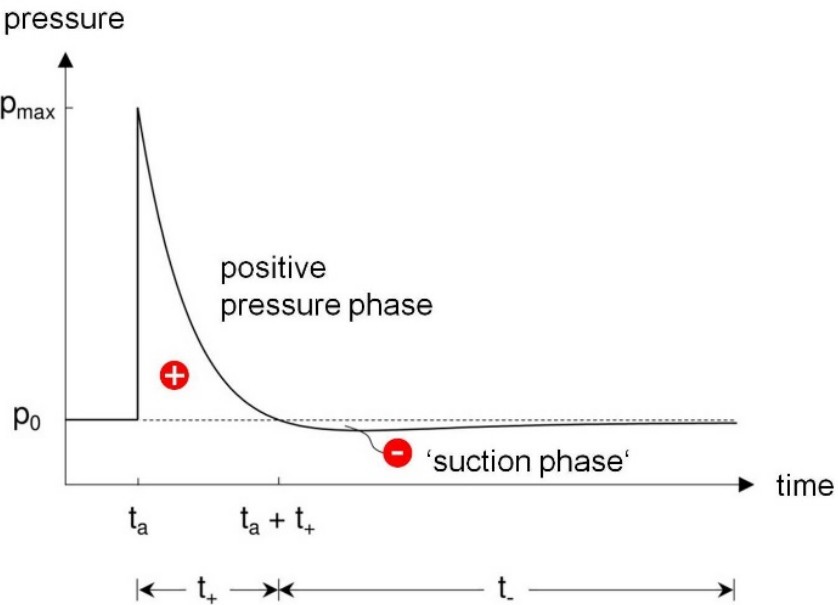

**Figure 11.** Temporal pressure curve (stationary) for detonation wave propagation in the free field according to the modified Friedlander equation.

In this framework, no consideration is given to possible amplification effects due to interactions between the pressure wave and the affected structure or the environment (ground/tunnel/nozzle effects). Depending on the strength of the incident wave and the angle of incidence, as well as the structure and its surrounding conditions, reflection processes and compression effects can lead locally to much greater overpressures than the original levels. In such cases, the pressure characteristic can be approximated in a simplified way via so-called reflection coefficients. Under ideal gas conditions, using coefficients of weight, with the ionization and dissociation of the air molecules in real gases taken into account, values of up to 20 have been documented [66].

The change in the momentum $I_+$ of the examined structure that is caused by the impacting pressure wave results from the duration of the positive pressure phase and the corresponding (scaled) pressure curve:

$$I_+ \ = \ \int_{t_+} p(t)\mathrm{d}t \tag{2}$$

The type of ambient fluid has a major influence, which is why—as described briefly below—sandwich constructions have inherent advantages over more massive monolithic plates of the same weight in underwater explosions. When hitting a plate (unsupported on the back), the pressure impulse sets it in motion and experiences a partial reflection. With the onset of tensile stresses ($\hat{=}$suction phase) at the contact surface between the fluid and the structure, the plate reaches its maximum velocity [67]. The effective change in momentum that is transferred to the component at this point $I_t$ can be calculated by means of the Taylor relationship [68]:

$$\frac{I_t}{I_0} \ = \ 2q^{\frac{q}{1-q}} \text{ with } q \ = \ \frac{t_0}{t_*} \tag{3}$$

Here, $I_0$ denotes the unreflected change in momentum of an ideal, one-dimensional pressure wave developing freely in space. Furthermore, the term

$$t_* \ = \ \frac{\rho h}{\rho_f c_f} \tag{4}$$

serves as a time scale factor characterizing the fluid–structure interaction. It strongly depends on the properties of the ambient fluid and the plate. Therefore, in Equation (4), $\rho$ and $h$ denote the density and thickness of the plate, respectively. $\rho_f$ is the density of the fluid on the impact side, and $c_f$ the speed of sound of the fluid. The decay time of the positive pressure phase is denoted by $t_0$. Figure 12 shows the momentum ratio as a function of $q$, as well as the representative temporal magnitudes for a homogeneous steel plate of thickness h = 0.0254 m. See also [67].

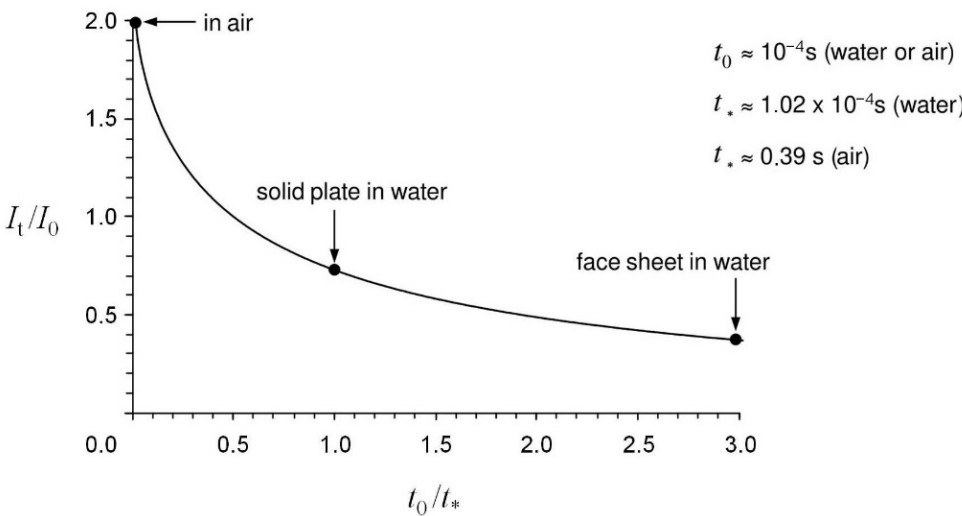

**Figure 12.** Plot of the momentum ratio $I_t/I_0$ as a function of the time scale ratio $q$ for assessing fluid–structure interaction according to the Taylor relationship.

Figure 12 provides simple interpretations for the two limits $t_0/t_* \ll 1$ and $t_0/t_* \gg 1$. Massive plates that are hardly set in motion by the (free-field) momentum $I_0$ ($t_0/t_* \ll 1$) generally ensure an almost perfect rebound of the impacting pressure wave ($-I_0$) so that the final force impact exerted on the plate is approximately $2I_0$. In general, the influence of the structure on the fluid movement decreases with decreasing plate thickness. In the limiting case of vanishing thickness, the plate acquires the velocity of the fluid and, like a free surface, experiences no momentum load. The significant effects of the ambient medium and the plate thickness on the transmitted momentum result from the specific time variable relationships. For a homogeneous plate in water, a value of $q = 0.98$ results in a transmitted momentum of about $0.75I_0$. In contrast, with the distribution of the total plate mass that is possible for sandwich construction plates with correspondingly thin cover layers, significantly reduced momentum loads can be achieved. For example, with a mass distribution of one-third each for the upper cover surface, the core, and the bottom cover surface, the corresponding cover surface is only hit by a force impact of approximately $0.38I_0$ ($t_* = 0.34 \rightarrow q \approx 3.0$). In the case of structures subjected to air-blast loading, this advantage is essentially non-existent due to the low density of the fluid. In that case, both a solid plate and a sandwich structure experience impulse changes of almost $2I_0$ since the times of decay and fluid–structure interaction differ by several orders of magnitude, and a change in the thickness ratios has no significant effect.

Deshpande and Fleck [69] employed a simplified sequential model to describe the typical structural response of a sandwich construction under explosive loading with three successive phases. Phase I involves the fluid–structure interaction already described above and the transfer of momentum $I_t$ to the upper surface layer. The resulting velocity

$$v_1 = \frac{I_t}{\rho h} \tag{5}$$

of the upper surface leads to the continuous compression and densification of the core (phase II). In the case of weak explosive loading, the densification front can be stopped within the core, and the velocity of the upper surface layer can be completely slowed down. In such a case, the kinetic energy is dissipated mainly by plastic deformations of the core structure. Larger impinging pulses ultimately cause the acceleration of the entire structure, with similar velocities at the upper and bottom sides (except for the edge areas). In phase III, the sandwich component bends, and additional dissipation processes occur due to plastic membrane distortions in the cover layers.

The spatial dependence of the pressure distribution due to different angles of incidence and propagation times of the shock wave remained mostly unconsidered in the previous explanations. However, such effects are included in the ConWep simulation method for explosion loads in air developed by the United States Army Corps of Engineers and implemented in ABAQUS/Explicit [70,71]. This empirical approach is based on extensive experimental data sets, and the model—after determining the ignition location and specifying the quantity of explosive in equivalent quantities of TNT—provides the maximum overpressure, the arrival time of the pressure front, and the duration of the positive pressure phase, as well as the decay coefficient both for the incident and for the reflected shock wave. By means of the analytical load function

$$P(t) = \begin{cases} P_i(t) \cdot \left[ 1 + \cos\Theta - 2\cos^2\Theta \right] + P_r(t) \cdot \cos^2\Theta, & \cos\Theta \geq 0 \\ P_i(t), & \cos\Theta < 0 \end{cases} \tag{6}$$

the resulting peak pressure on the corresponding element nodes can be determined if information about the distance to the detonation source, the amount of charge, and the orientation of the affected structural components is given. In Equation (6), $P_i(t)$ denotes the incoming pressure, $P_r(t)$ is the reflected pressure, and $\Theta$ the angle of incidence, which is defined as the angle between the normal of the loaded surface and the vector directed from the loaded surface to the ignition source. The model includes empirical parameters for detonation processes near the ground with hemispherical wave propagation, as well as for ignition locations at higher altitudes with unhindered spherical pressure wave expansion. The ConWep approach reproduces the pressure–time relationships realistically and removes the need for the simulation application to model the surrounding medium (a pure Lagrangian consideration), which results in a considerable reduction in the computational effort. Only fluid–structure interactions, such as shadowing and amplification effects due to any obstacles, cannot be included.

## 5. Description of the Numerical Model

In this section, a detailed description of the underlying numerical model is provided. Figure 13 illustrates the support configuration and some of the constant geometry parameters. This setting is valid for all the topologies and configurations investigated here. The set of constant parameters also includes a constant cover thickness of 1.0 mm, a constant core height of 20.0 mm, and a constant relative core density of almost 3.3 percent. The outer dimensions of the sandwich specimens are 400 × 400 mm and do not vary. In all computational studies, both the total mass and the ratio of the core to the two faces (≈1:3) are kept constant. The core mass is thus approx. 25% of the total weight. In consequence, these specifications imply, for a monolithic reference panel of the same weight, a panel thickness of 2.66 mm. All components have the material properties of the aluminum alloy EN AW-7108 T6.

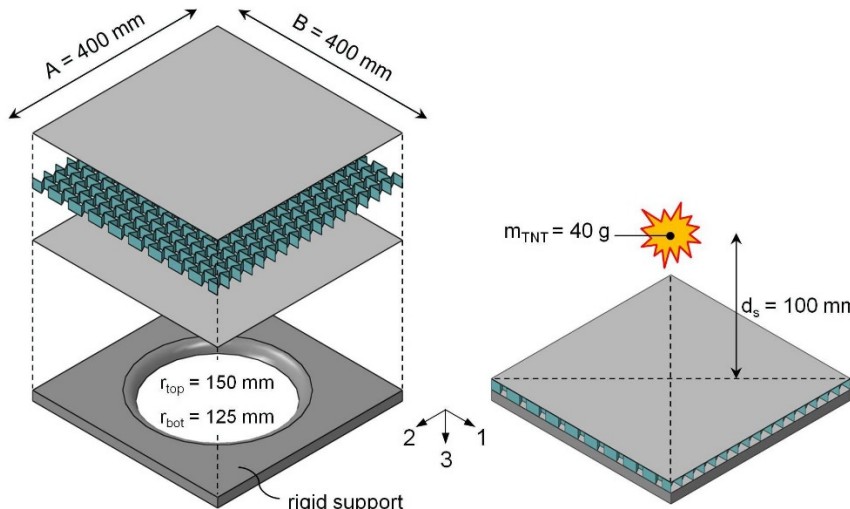

**Figure 13.** Configuration of the numerical model, including the support structure and the load source (air blast).

The aluminum alloy EN AW-7108 T6 serves as an exemplary representative for descriptions of ductile material behavior. The main alloying element, zinc, helps it to achieve high strengths and allows it to be used in aircraft and automotive construction, aerospace applications, and high-quality sports equipment. The alloy has high corrosion resistance, favorable machining properties, and very good weldability. It is, therefore, a suitable material for the real production of the different core variants for sandwich panels. The elastic material parameters and the density of the aluminum alloy are shown in Table 5.

**Table 5.** The elastic material parameters and the density of the aluminum alloy EN AW-7108 T6.

| Young's Modulus | Poisson's Ratio | Density |
| --- | --- | --- |
| 70.000 N/mm$^2$ | 0.33 | 2.700 kg/m$^3$ |

The commercial finite element code ABAQUS/Explicit employs an extensive material model that includes rate-dependent elasto-plastic material behavior and various types of failure mechanisms. The calculations for damage evolution and progressive softening after damage initiation are based on the concept of effective stresses. The behavior of the damaged material is represented by the behavior of the undamaged material in conjunction with a total scalar damage parameter D. This parameter takes into account the individual stresses specified for the three ductile failure mechanisms:

- Failure due to the growth, coalescence, and regeneration of micro-voids.
- Shear/slip failure due to local shear bands.
- Necking instabilities (local necking).

By means of the damage parameter, both reductions in the yield stress and losses of stiffness can be modeled. All necessary material data, including, in particular, those for characterizing the hardening and strain rate behavior as well as the failure phenomena, are taken from the experimentally verified ABAQUS reference example "Progressive failure analysis of thin-wall aluminum extrusion under quasi-static and dynamic loads".

A blast loading above the support structure serves as the load source. Measured from each upper edge, the point of ignition is located at a distance of 100 mm centrally above the alternating test structures (i.e., the different core topologies). The TNT equivalent of the explosive charge is 40 g. The support is free from any kind of constraints in the sense that the sandwich structure lies loosely and directly on the supporting structure and is free in its translational and rotational movements throughout the simulation. A restriction of movement exists only due to the contact conditions. The ABAQUS/Explicit general

contact conditions prevail throughout the model (hard contact in the normal direction and "aluminum–steel" or "aluminum–friction" contact in the tangential direction). However, all components of the sandwich structure can come into mutual contact. Besides this internal contact, interaction between the overall sandwich structure and the support is also ensured. The support is modeled as a rigid body whose degrees of freedom are locked in all directions.

The cover surfaces are modeled exclusively with so-called S4R shell elements, see Figure 14. A meshing strategy that is decoupled from the core ensures uniformly structured top and bottom surface meshes that are identical for all variants. The mesh size is one millimeter. The meshing of the core structure, on the other hand, varies with the geometry of each core. In principle, S4R elements of the same order of magnitude are also used for all prismatic core topologies that involve a structured mesh. The element type only changes in the case of 3D lattice cores. For this type of core topology, spatial Timoshenko beam elements (B31) represent both the auxetic and the non-auxetic cell-like core structures. Investigations by Deshpande et al. [72] showed very good agreement of the numerical data for this modeling approach. The ABAQUS tie constraint guarantees an ideal connection between the individual sandwich components at the respective interface points.

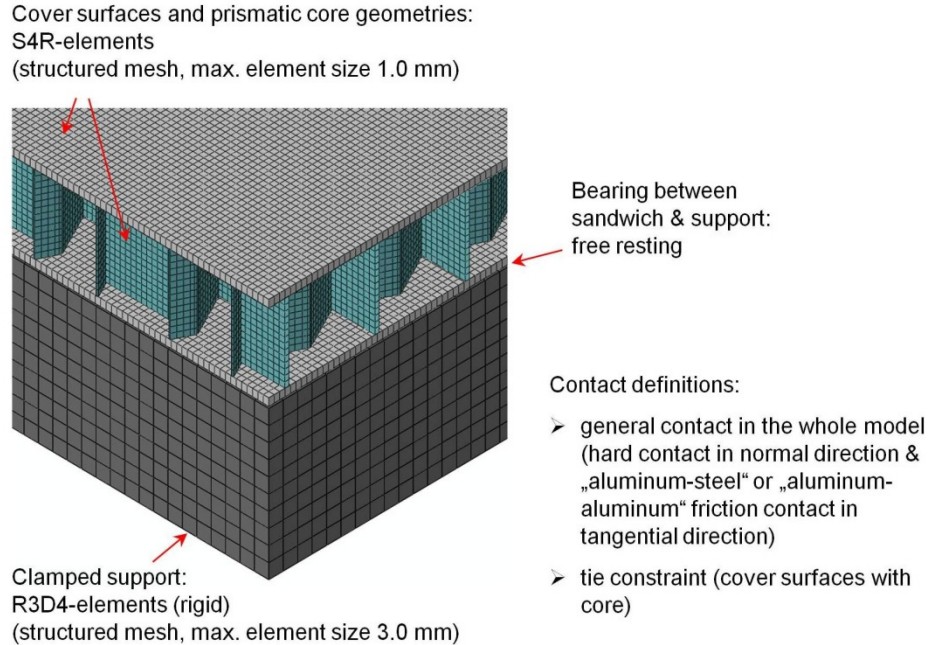

**Figure 14.** Exemplary presentation of the FE-mesh as well as bearing and contact conditions.

## 6. Simulation Results

The investigation of the non-linear structural behavior of several types of sandwich construction subjected to the given blast loading provides the basis for an assessment of the protection potential of the selected core topologies. Details of the simulation results are summarized in the following charts. These charts contain the temporal evolution of the protection-relevant mechanical parameters, such as the displacements and accelerations that occurred on the top and bottom sides in the center of the panel, as well as the reaction forces that were observed at the supporting structure ($RF_3$) and the plastically dissipated energy (ALLPD) in the time interval under consideration. In addition, a comparative compilation of the respective maximum values–which are normalized by individually dividing them by the corresponding reference values for the monolithic reference plate of the same area and mass–enables a clear qualitative classification of the different design strategies. The aforementioned values represent typical mechanical parameters useful for assessing structural performance and estimating possible physical injury in dynamic stress situations.

For purposes of illustration, the discrete evaluation points are shown in Figure 15. At the reference nodes, the respective displacements of the cover surfaces in the center of the field ($u_{3,\text{top}}$ and $u_{3,\text{bot}}$) and the corresponding accelerations ($a_{3,\text{top}}$ and $a_{3,\text{bot}}$) were evaluated. For the homogeneous monolithic reference plate, the two evaluation points on the shell center surface coincide so that it holds $u_{3,\text{top}} = u_{3,\text{bot}}$ and $a_{3,\text{top}} = a_{3,\text{bot}}$.

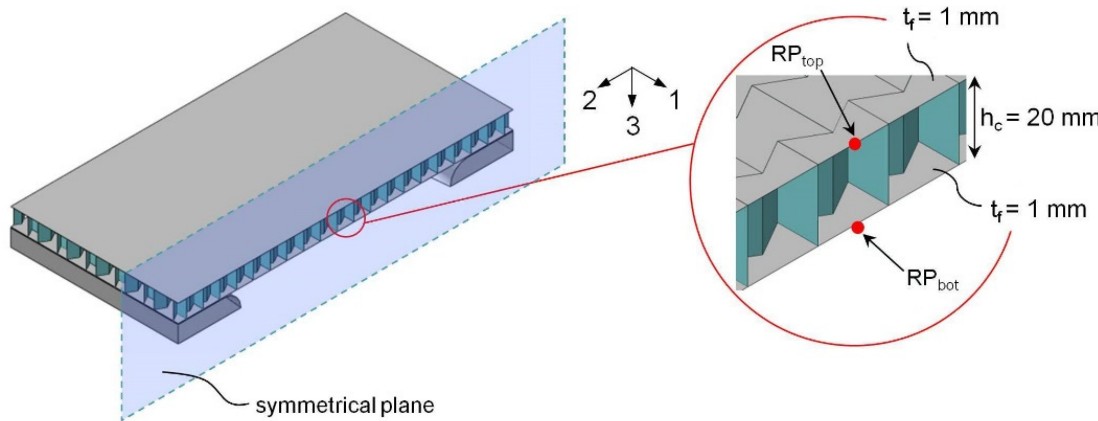

**Figure 15.** Position of the discrete evaluation points in the center of the panel on the upper and lower cover surfaces.

In addition to the measurable local and integral mechanical variables, the shape of the global (system) deformation also showed considerable differences for the investigated sandwich variants. Structural integrity can, however, play an important role in the application of protective devices, depending on the field of application and requirement profile. Figure 16 shows the deformation of the monolithic reference panel as a result of the specified explosion load. The largest deflections were observed at the measuring points in the center of the panel (RP$_{\text{top}}$ and RP$_{\text{bot}}$).

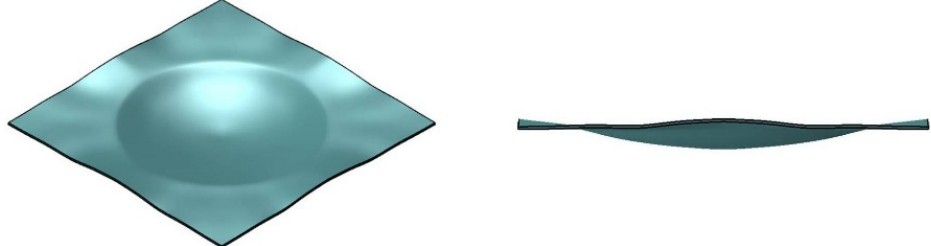

**Figure 16.** Deformation of the monolithic reference plate under blast loading (t = 0.6 ms).

In contrast, various sandwich structures—in some cases, despite good mechanical characteristic values—exhibited considerable deformations and failure states. Some exemplary deformation states for all three classes of core topology are shown in Figure 17. A more comprehensive overview can be found in Appendix A. The simulations show that the group of honeycomb cores was, without exception, the most resistant and robust sandwich type. For honeycomb cores, the majority of the deformations occurred in the unsupported area, whereas the supported part remained almost unchanged even after loading. Both deformations in the core area and global structure deformations were comparatively minor. None of the considered honeycomb core sandwich structures showed cracks in the cover surfaces throughout the finite element simulations.

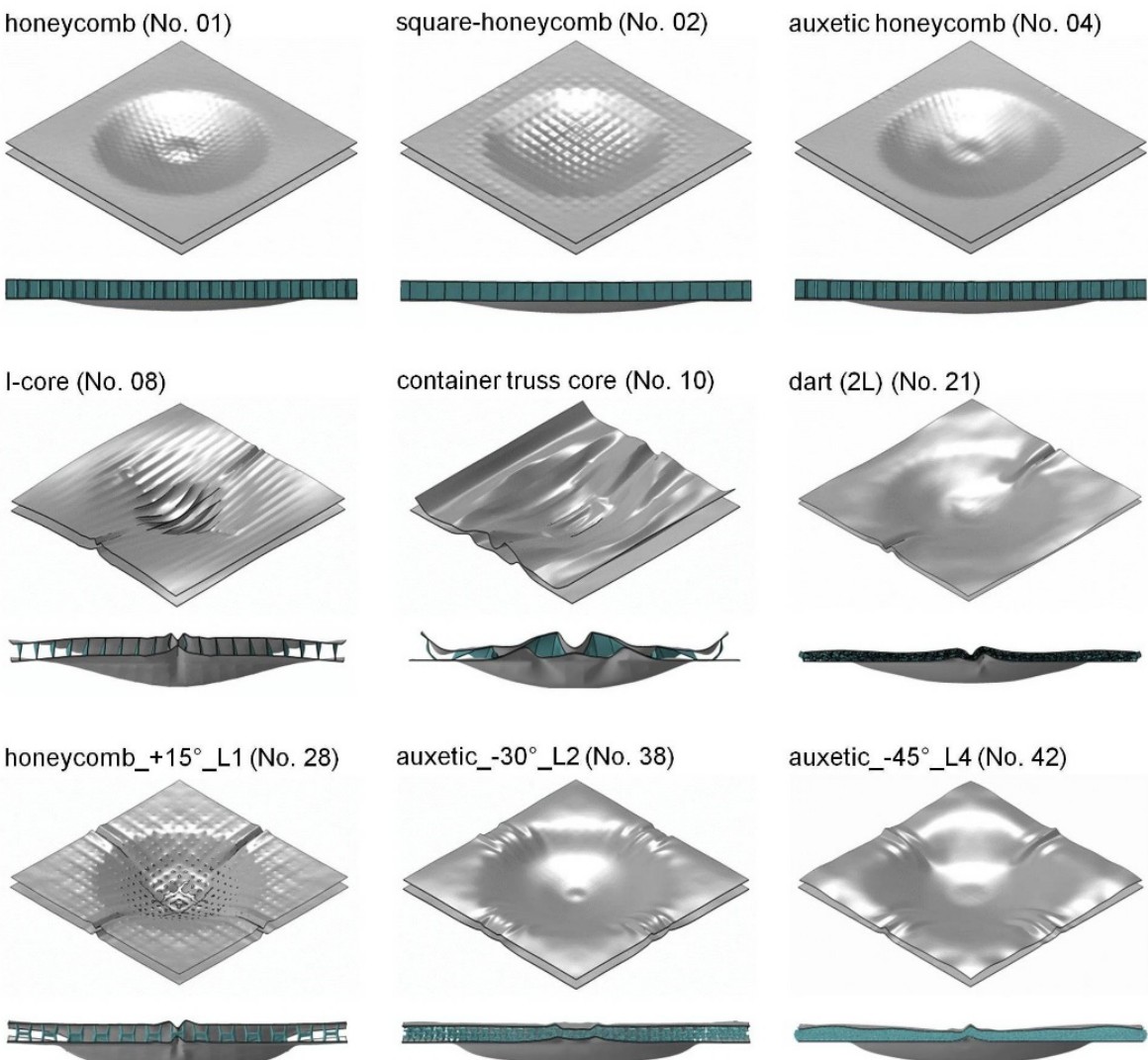

**Figure 17.** Deformation states of selected sandwich variants under blast loading (t = 0.6 ms).

For the 3D lattice cores, in addition to their dependence on the lattice cell angles, the strong influence of the number of layers on deformation behavior was observed. The existence of few support points in the one-layer 3D lattice core designs led, with positive or neutral cell angles, to the perforation of the top surface.

On the other hand, auxetic cells prevented perforation and cracking even in one-layer designs due to the build-up of additional compressive stresses in the upper cover surface during deformation. An increasing number of layers tended to reduce the occurrence of failures, irrespective of the cell angle.

Figure 18 summarizes the temporal evolution of the relevant mechanical parameters of one representative for each core type; the respective reference values of the monolithic plate are also included. These values are later used for normalization. Apart from the dissipated energy, all quantities reached their absolute maximum within 0.6 milliseconds. This applies to all core variants considered in the article. Both the substantial performance capacity of these variants and the considerable performance variations among the different core designs for sandwich structures are evident even from this first overview.

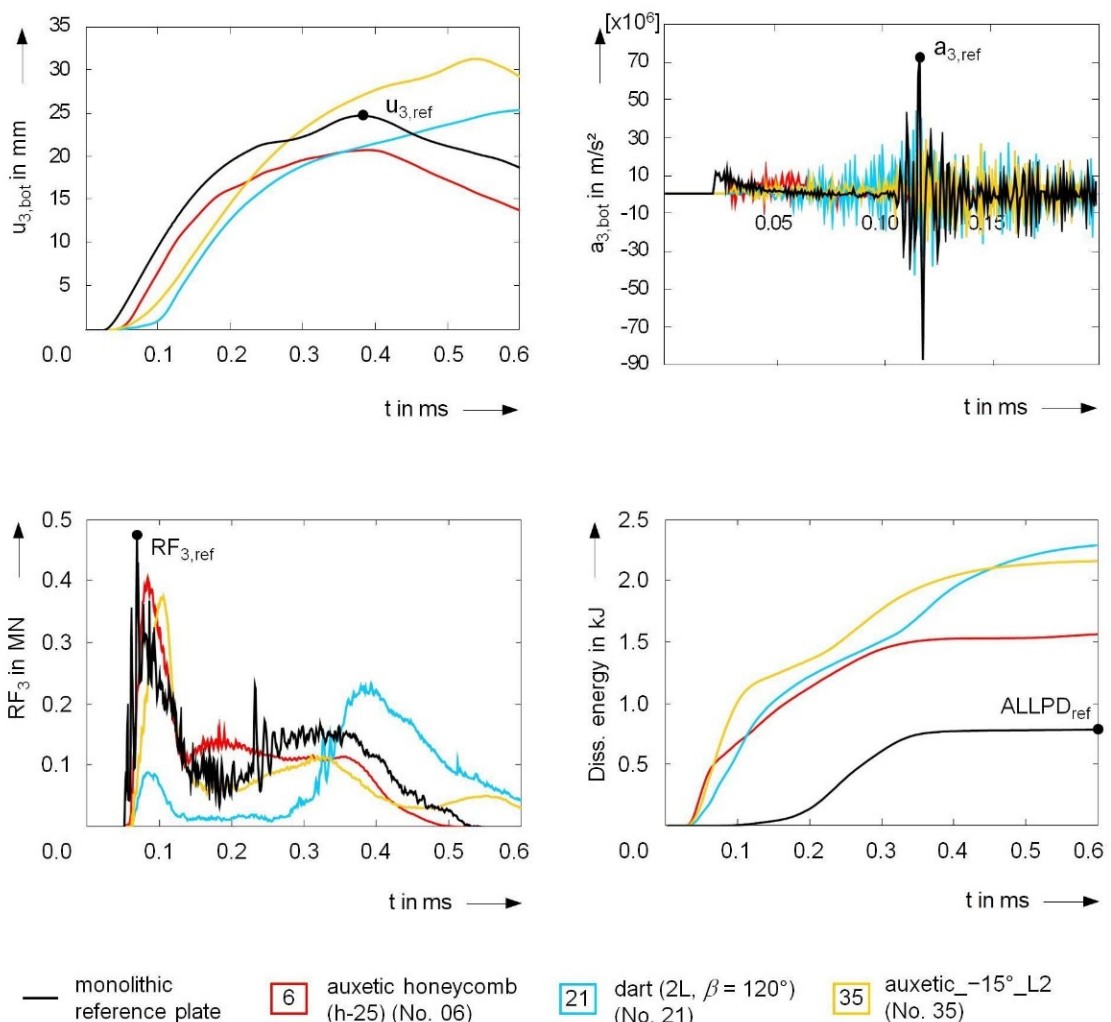

**Figure 18.** Temporal evolution of the relevant mechanical parameters during blast loading.

The qualitative and quantitative displacement curves do not present a uniform picture. For both the prismatic and the lattice cores, the maximum displacements of the lower cover surfaces occurred significantly later than for the homogeneous reference plate, and these values were also above the reference value of almost 25 mm. The selected auxetic honeycomb core, on the other hand, exhibited a temporal behavior comparable to the reference, with a reduced maximum deflection of about 20 mm. This also corresponded to the development of the energy dissipated by plastic deformations. As the main global deformations subsided and the local maximum displacements were reached, the energy dissipation practically reached individual plateaus. Due to the clearly pronounced deformations in the core area, the dissipation curves of the sandwich components drastically exceeded the reference characteristic curve of the monolithic reference panel from the very start of loading. In addition, in the case of the reaction forces transmitted to the substructure, the timings of the highest occurring load varied considerably. However, a reduction in the maximum forces was observed throughout all core designs. The reference value of approximately 0,5 MN was undercut considerably in some cases.

A similar result was obtained when evaluating the accelerations. The measuring points experienced frequent changes in acceleration direction. Since the greatest interest in practical applications is the protection of the opposite side of the load, (positive) accelerations in the positive z-direction are considered consistently throughout this article. Here, the use of sandwich structures of the same mass led to a significant reduction in the acceleration peaks at the rear cover surface in the center of the panel.

Since the selected examples provide insight into the spectrum of qualitative curves and absolute values, only the core-dependent changes in maximum values compared to the monolithic reference plate are of interest in the following examples. First, let us examine the group of honeycomb cores. Due to the dominating membrane stress of the core constructions in the examined loading situation, this type of sandwich core design exhibited very stiff behavior. The deflection of the bottom face was reduced by up to 25% compared to the monolithic reference panel by using a conventional regular hexagonal construction; see Figure 19.

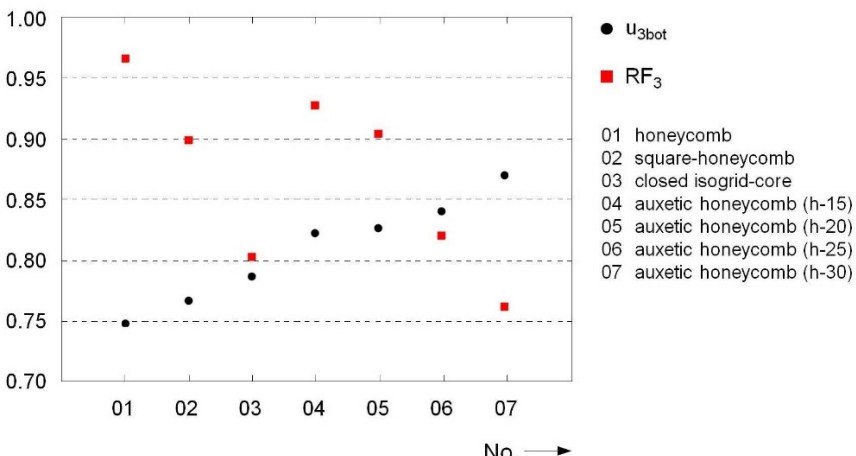

**Figure 19.** Honeycomb cores: normalized maximum values for displacement $u_{3,\text{bot}}$ and reaction force $\text{RF}_3$.

The displacements in the auxetic cores increased steadily with increases in honeycomb size; values of 12.5–17.5% below the reference value were observed. A qualitatively similar relationship was observed for the sandwich top cover, as seen in Figure 20. With increases in "span width" within the individual auxetic cells, the deformation also increased and finally reached values above the reference value. Despite its larger cell support spans, the honeycomb pattern tended to have slightly smaller deflections. The maximum values of the other topologies were in between for both evaluation points.

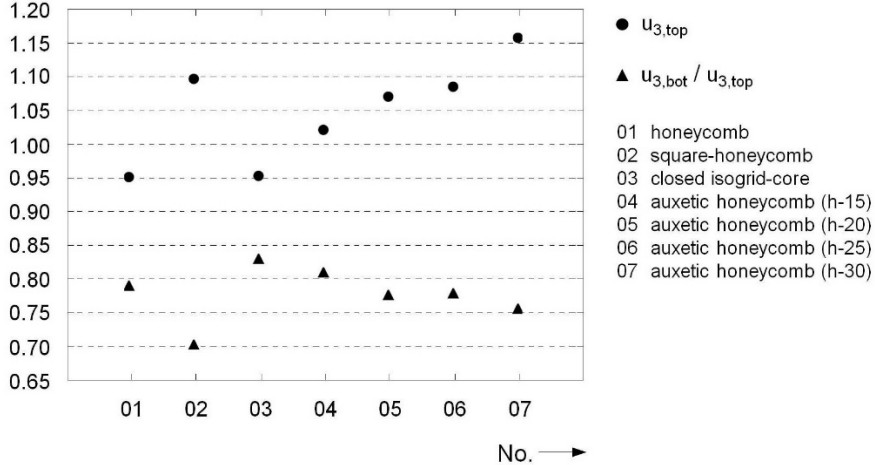

**Figure 20.** Honeycomb cores: normalized maximum values for displacement $u_{3,\text{top}}$ and displacement ratio $u_{3,\text{top}}/u_{3,\text{bot}}$.

The comparatively high buckling resistance of honeycomb core cell walls was also reflected in the ratio of lower to upper displacement. Here, all considered core configurations were in the range of 75–80%, as seen in Figure 20. For this reason, the opposing

cover surfaces retained most of their initial distance (see also Appendix A). Only the sandwich construction with square honeycombs displayed a somewhat softer behavior. In the auxetic variants, larger cell support widths tended to lead to increases in compliance and deformations within the core area. The development of the dissipated energy supports this statement, as shown in Figure 21. Here, a clear increase with increases in cell size was observed. In total, almost twice as much energy was dissipated in all topologies compared to the reference plate during the entire observation period. Due to its robust core structure, the honeycomb exhibited the lowest dissipation capacity, with a factor of 1.85. With the conventional hexagonal honeycomb, this also led to an almost undiminished pulse transmission and peak load on the substructure; see Figure 19. The use of auxetic cells, on the other hand, reduced the maximum occurring load by up to 25%. With a load reduction of about 20%, the isogrid core also had an advantage over the monolithic panel.

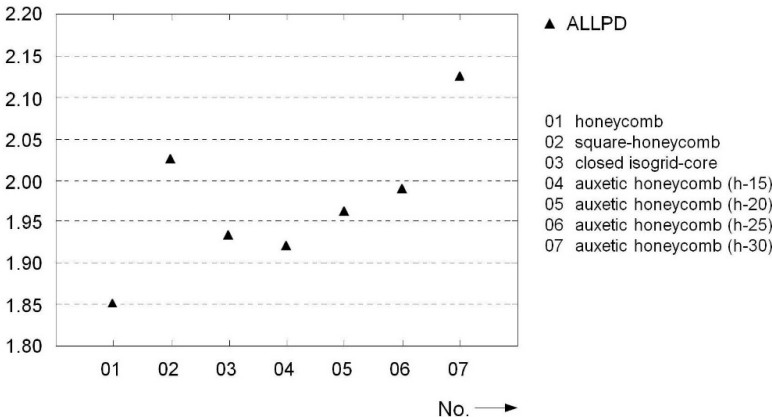

**Figure 21.** Honeycomb cores: normalized maximum values for plastic dissipation energy.

Figure 22 illustrates at a glance the type-related differences between the unilaterally open prismatic cores and the previous honeycomb designs. The displacements of the bottom cover surfaces all exceeded the reference value, including, in one case, by up to 40%. Only the isogrid and two auxetic variants (No. 19 and No. 21) were on par with the monolithic reference panel. The displacements of the upper surface were also much more pronounced and, in some cases, more than twice that of the reference plate; see Figure 23. Both the non-auxetic (No. 8–17) and the prismatic variants (No. 18–21), in contrast to the honeycomb cores, were primarily subjected to bending stress under the given load. This is evident in the reduced displacement ratio of the lower to the upper surface. In principle, this type of core structure is rather soft and exhibits considerable deformations as well as plastic changes in shape.

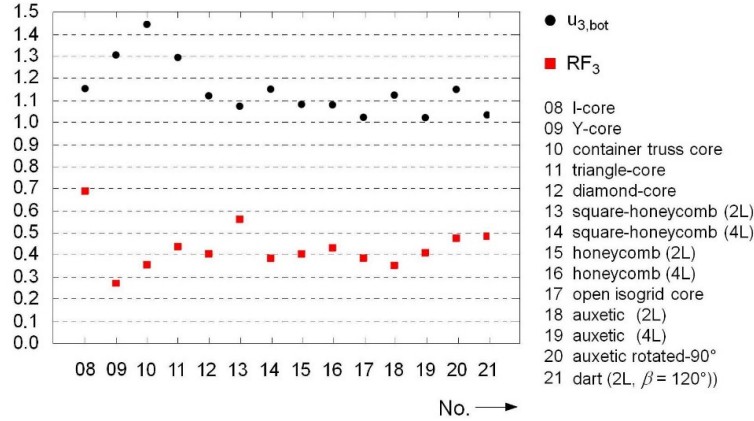

**Figure 22.** Prismatic cores: normalized maximum values for displacement $u_{3,\text{bot}}$ and reaction force $RF_3$.

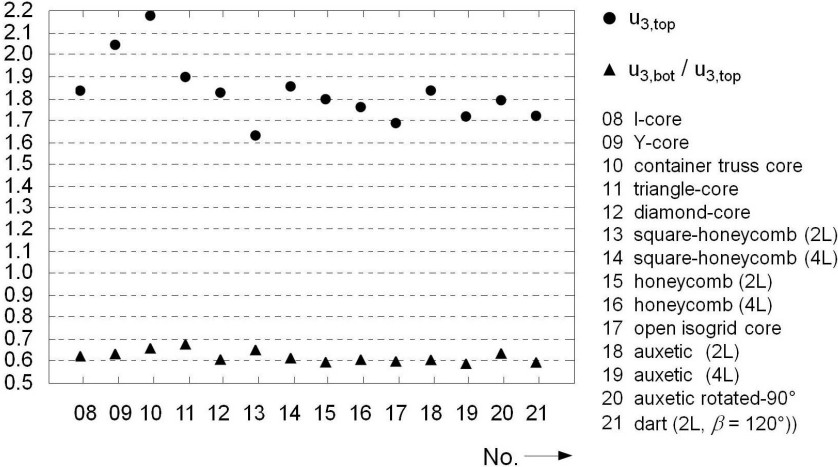

**Figure 23.** Prismatic cores: normalized maximum values for displacement $u_{3,\text{top}}$ and displacement ratio $u_{3,\text{top}}/u_{3,\text{bot}}$.

The plastic energy dissipations are correspondingly higher. With factors between 2.8 and 3.1, the prismatic variants not only surpassed the performance of the reference panel but also that of the sandwich components with honeycomb cores; see Figure 24. The geometries with the highest residual membrane stiffnesses (I-core and square honeycomb 2L/4L) showed significantly lower levels of plastic deformation and, consequently, dissipated less energy. Apart from these two geometries, the considered prismatic core topologies reduced the peak load transmitted to the support structure by about 60% on average, with the Y-core structure reducing it by more than 70%; see Figure 22.

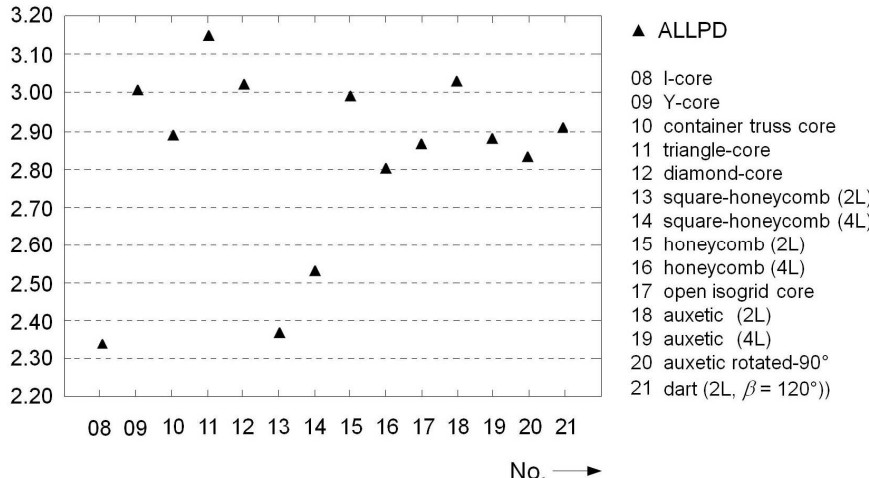

**Figure 24.** Prismatic cores: normalized maximum values for plastic dissipation energy.

With regard to the desired mechanical properties, i.e., low levels of deformation and acceleration on the side facing away from the load, a conflict of objectives arises; lower stress on the adjacent components simultaneously ensures a higher energy dissipation capacity. Obviously, core topologies that are predominantly subjected to bending stress tend to display a significant reduction in the resulting load on the adjacent components. However, at the same time, these cores exhibit comparatively high levels of deformation of the bottom side. Please note that the results for the sandwich structure with a four-layered auxetic pattern (No. 19) do not match this observation. In this study, the deflections on the bottom cover surface did not differ from those of the monolithic panel of the same mass. At the same time, the maximum load on the bearing structure was only 40% of the reference value.

Conventional and auxetic topologies from the class of three-dimensional lattice cores concluded the numerical study. As already described above, the auxetic variants were able to counteract the formation of cracks even in a single-layer design due to their unusual deformation behavior and the associated reduction in tensile stress in the upper surface areas. At the same time, this led to a reduction in upper displacement compared to the non-auxetic lattice geometries; see Figure 25.

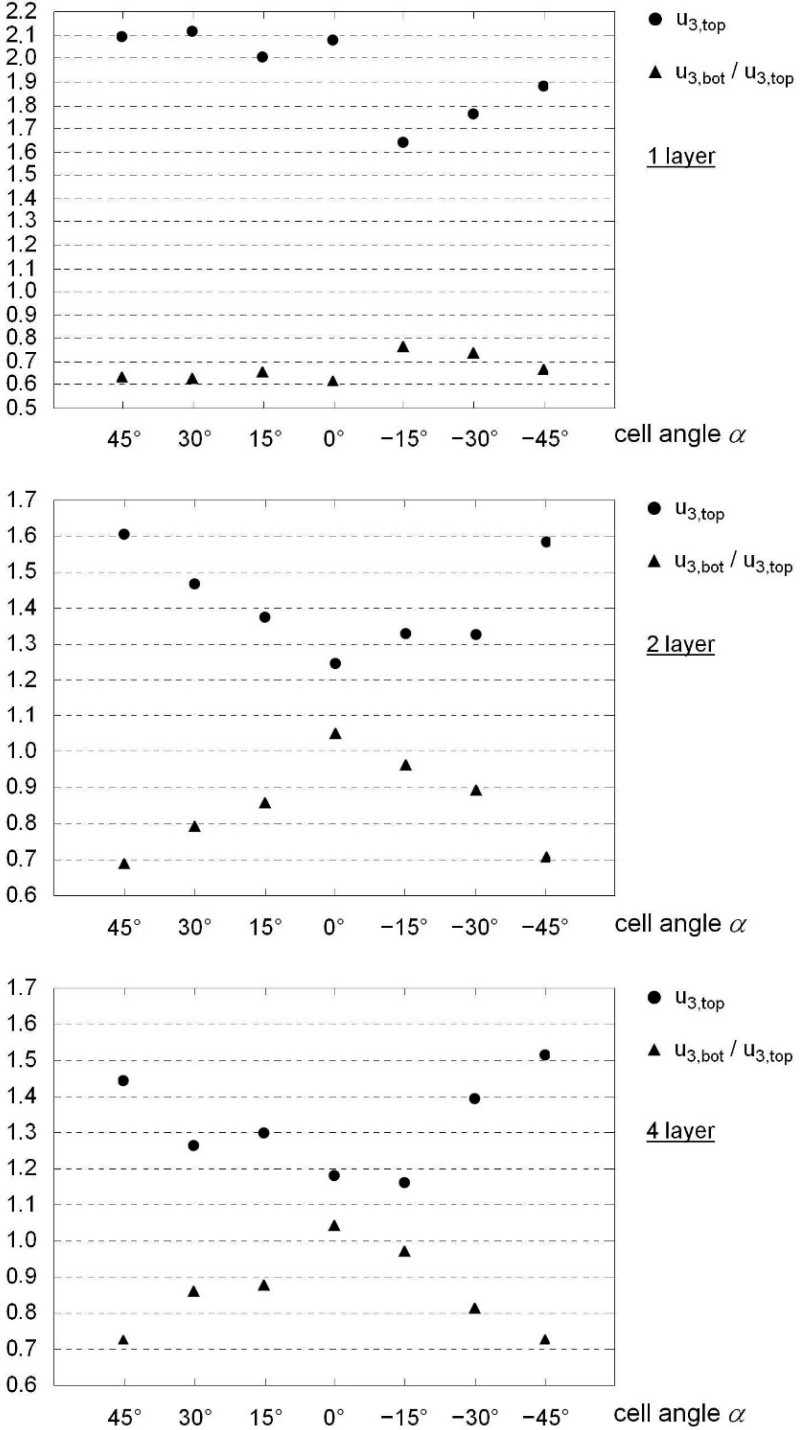

**Figure 25.** Three-dimensional lattice cores: normalized maximum values for displacement $u_{3,\text{top}}$ and displacement ratio $u_{3,\text{top}}/u_{3,\text{bot}}$.

However, this effect was not accompanied by a significant reduction in the deflection of the lower cover surfaces. Irrespective of the cell angle, the displacement was, on average, about 30% greater than in the reference plate; see Figure 26. While both the conventional single-layer lattice cores and the prismatic cores exhibited rather soft behavior due to partial failure, the auxetic lattices achieved higher core stiffnesses. In some cases, their stiffness was comparable to that of the honeycomb cores. Due to their general position in space, a mixture of bending and membrane stresses, without the clear dominance of one type, was always present in the lattice bars, even if the cell angles were adjusted.

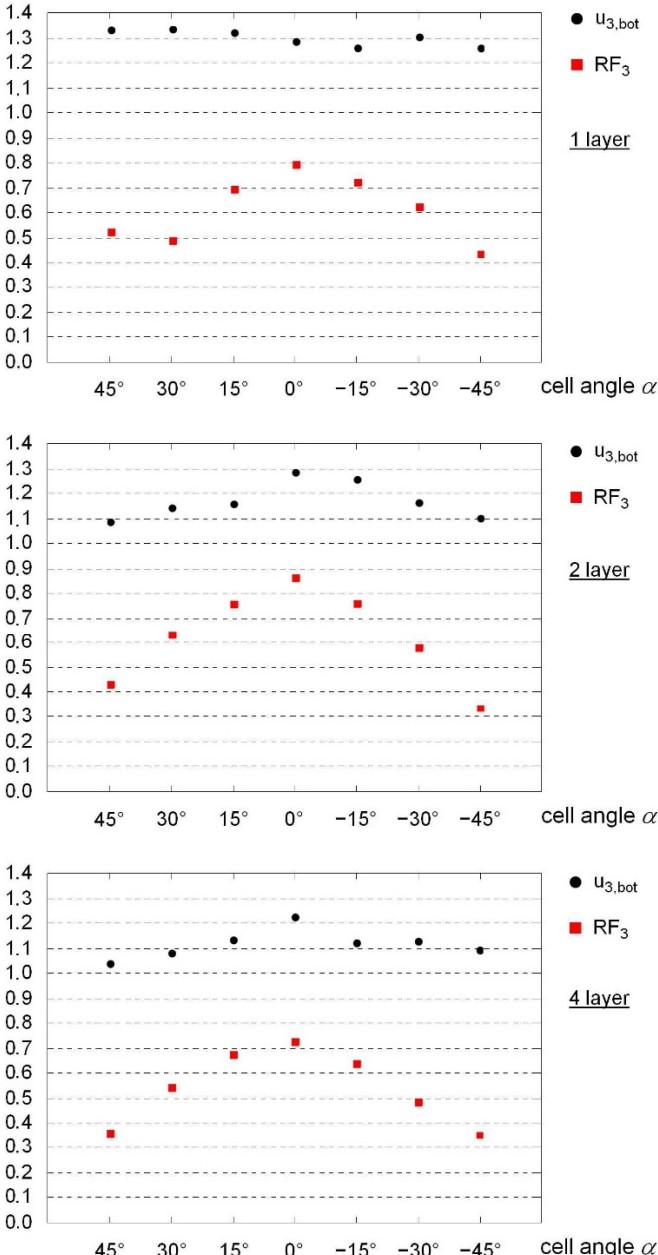

**Figure 26.** Three-dimensional lattice cores: normalized maximum values for displacement $u_{3,bot}$ and reaction force $RF_3$.

In general, the values for plastic energy dissipation capacity were quite high, as shown in Figure 27. Irrespective of the sign, it tended to increase with increases in cell angle. The opposite was true for the transmitted peak load. Here, the maximum stress on the substructure decreased considerably with increases in cell angles and dropped by up

to almost 60% compared to the homogeneous monolithic reference plate. Clear trends were observed with increases in the number of layers. The absolute values for deflection, both on the bottom side and, above all, on the top surface, were reduced in multi-layer structures. Larger cell angles led to deformations on the side facing away from the load, with a simultaneous increase in deformations on the top side. The values for the bottom surface were, in some cases, only slightly above the reference value of the monolithic plate. The stiffness of the core structure (in terms of the displacement ratio between the top and bottom sides) varied by up to 35% according to the cell angle. In this case, larger angles led to larger deformations in the core area, as well as increasing energy dissipation capacities. It is noticeable that with neutral cube cells ($\alpha = 0°$), very stiff cores were created that, despite the given compressive load, expanded in some areas of the core structure in the thickness direction. A strong angular dependence, with significant internal differences of up to almost 60%, also occurred in the case of maximum load transmitted. Here again, larger angles caused a drastic reduction in the transmitted load. The two-layer auxetic configuration ($\alpha = -45°$), for example, reduced the ambient stress by almost 70% compared to the reference value; see Figure 26.

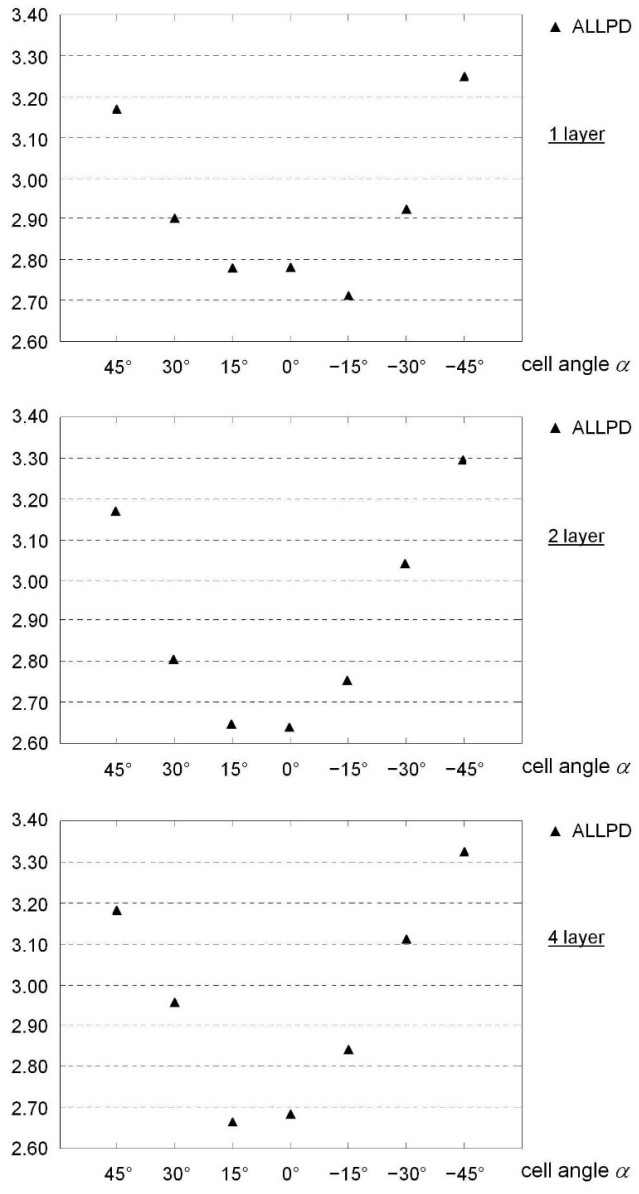

**Figure 27.** Three-dimensional lattice cores: normalized maximum values for plastic dissipation energy.

## 7. Discussion

The focus of this contribution, in contrast to most of the existing literature, is to conduct an assessment of fundamental—especially auxetic—topological influences while keeping other variables constant, such as total weight, relative core density, core height, outer dimensions, cover thicknesses, etc. The results confirm and extend the previously existing knowledge in the literature on the dynamic structural behavior of sandwich constructions under explosive loading (air blast).

Depending on the configuration, some of the studied core designs significantly improved performance in terms of plastically dissipated energy, maximum displacement values, and peak load on adjacent structures compared to homogeneous monolithic plates of the same dimension and mass. The details of the dynamic structural response to the blast loading, as well as the local and global failure processes, strongly depended on the distribution of mass over the thickness and the surface.

The class of honeycomb cores exhibited the lowest level of deflection overall on the bottom side, and all values were well below the values of the reference panel. Honeycomb cores are characterized by an extremely stiff core structure with limited internal deformation states. Accordingly, plastic energy dissipation tended to be low compared to the other core types. Among this class, there was a tendency for the auxetic honeycomb core variants to have a higher level of deflection. Due to the geometry-related negative system transverse strain, the natural deformation pattern of an auxetic core when subjected to bending stress led to a convex, equidirectionally curved dome shape. The conventional non-auxetic pattern, on the other hand, tended to form a saddle-shaped surface under load. Obviously, honeycomb structures have a slightly higher bending or system stiffness, which manifested itself in somewhat lower levels of displacement under the considered shock wave loading. At the same time, the more flexible auxetic behavior was associated with higher levels of energy dissipation and a reduction in the transmitted forces.

In the group of prismatic core structures, completely different load-bearing behavior was prevalent in the majority of the topologies. From the beginning of the loading process, the structural behavior inside the core was primarily dominated by bending stresses on the cell walls, which was successively accompanied by significantly reduced cover surface distances and reduced system stiffnesses. As a direct consequence of the more flexible core structure, generally higher and, in some cases, considerably higher levels of deformation resulted at the upper and lower cover surfaces compared to both the honeycomb cores and the reference panel. In contrast, there was a considerable reduction in the peak load transmitted to the substructure: up to a 70% reduction compared to the reference panel. At the same time, the energy dissipation capacity was significantly increased due to the strong plasticization of the core structure.

Since a varying lattice cell angle is an elementary geometry parameter of three-dimensional lattice cores, no primary load transfer mechanism can be assigned to them. Rather, with their basic topology, the inclination angles of the lattice bars determine whether there is locally a dominant bending or membrane load or a balanced, mixed form. As the lattice cell angles increase in magnitude, the bending influence on the inclined bars increases as well; however, at the same time, the bars running parallel to the cover surfaces experience an increasingly higher axial load. Here, it does not matter whether the design is auxetic or non-auxetic. In both cases, larger cell angles reduce the deflection of the lower surface and, at the same time, they reduce the transmitted forces. This is an extremely desirable relationship between these two elementary structural parameters, which has not been observed to the same extent in the previously considered honeycomb cores or in the prismatic core geometries. A fundamental advantage of auxetic lattice cores over conventional 3D lattice cores results, once again, from their typical deformation behavior. The auxetic tendency towards contracting the lateral movements of the lattice bars under compressive loading led, in the present case, to a positive change in the stress state during the explosion loading, especially in the upper cover surface. Locally reduced tensile stresses prevented the formation of cracks in the sandwich surface within the framework of the

failure criteria employed in this study. In contrast to the results for the non-auxetic counterpart, the core topologies of the auxetic variety prevented the occurrence of structural failures even with a single-layer design.

For a more compact assessment of all core types and their sub-variants, it is necessary to determine a suitable scalar evaluation measure from the numerous protection-relevant structural parameters. Decisive criteria for the quality and serviceability of a sandwich construction might be, for example, the deflection of the side facing away from the load and the forces distributed onto neighboring structures. Depending on the field of application, the focus could also be put on other structural features. In the context of this study, a possible performance indicator is given by the product of the normalized maximum displacement of the bottom cover surface ($u_{3,\text{bot}}$) and the normalized peak load of the substructure ($RF_3$). Thus, the reference configuration has a coefficient of performance of 1.0. Lower values indicate an improvement in the overall structural behavior.

The results obtained using the abovementioned indicator are shown in Figures 28 and 29. Due to the relatively high values of around 0.7 for the honeycomb cores, the indicator reveals at first glance a conflict between high (dynamic) stiffness properties and low momentum transfer. Although suitable auxetic configurations can improve the performance potential of conventional honeycomb geometry, the significantly lower reaction forces are associated with comparatively higher levels of deflection. The indicator is considerably lower for the prismatic core constructions due to reduced force transmission. The Y-core and some auxetic variants achieved indicator values of less than 0.4. The largest fluctuations in the performance spectrum are observed for the three-dimensional lattice cores, with a range of values between 0.35 and 1.1. Here, the configurations with the largest angles in terms of magnitude, irrespective of the number of layers, had the best efficiency values. Due to their lower susceptibility to failure, auxetic lattice topologies represent an interesting design alternative with high potential for improvement.

At this point, the transfer and generalization of the obtained results to comparable systems and loading situations remain open possibilities. Within the scope of this work, it cannot be clarified whether qualitatively comparable structural characteristics can be achieved with changes to external dimensions and shapes (one or more curved sandwich components), variations in relative core densities, higher load intensities, material substitutions, etc.

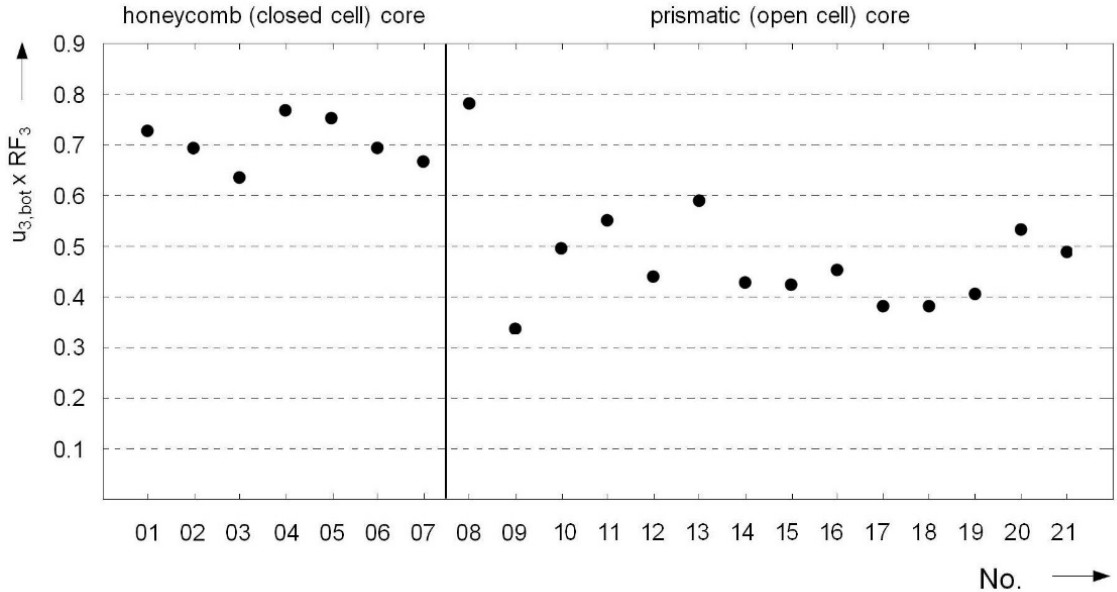

**Figure 28.** Performance indicator for honeycomb (closed-cell) cores and prismatic (open-cell) cores.

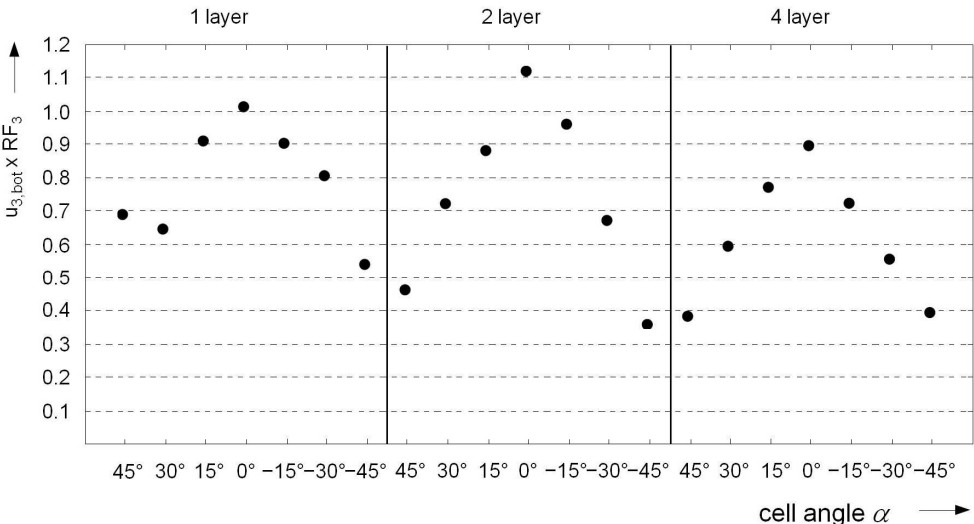

**Figure 29.** Performance indicator for 3D lattice cores.

**Author Contributions:** Conceptualization: all; methodology: all; formal analysis, implementation, evaluation of simulation results, investigation, and validation: M.W. and U.R.; resources, M.W. and D.A.; writing—original draft preparation, D.A.; writing—review and editing, all; visualization, M.W. All authors have read and agreed to the published version of the manuscript.

**Funding:** This research received no external funding.

**Institutional Review Board Statement:** Not applicable.

**Informed Consent Statement:** Not applicable.

**Data Availability Statement:** The data presented in this study are available on request from the corresponding author.

**Conflicts of Interest:** The authors declare no conflict of interest.

## Appendix A

In this appendix, supplementary results from the numerical investigations for the selected core variants are listed. The simulation results provide a good overview of the characteristic deformation states of the various core topologies.

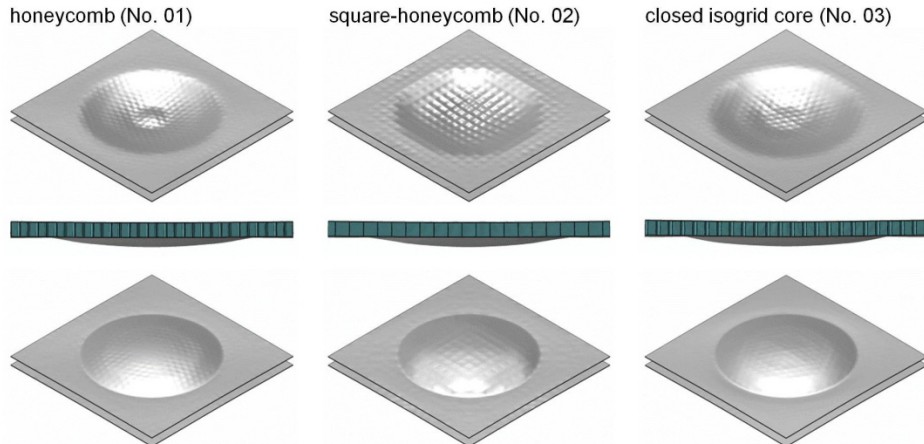

**Figure A1.** Deformation states of the top and bottom sides for honeycomb cores No. 01–03 subjected to air-blast loading (t = 0.6 ms).

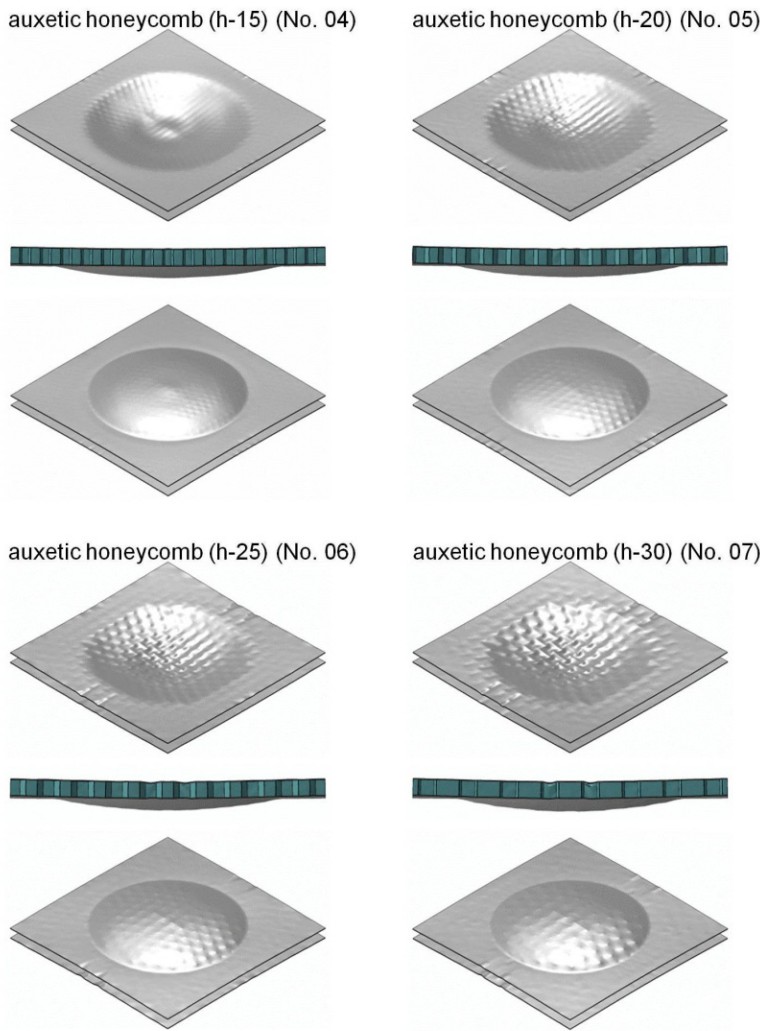

**Figure A2.** Deformation states of the top and bottom sides for honeycomb cores No. 04–07 subjected to air-blast loading (t = 0.6 ms).

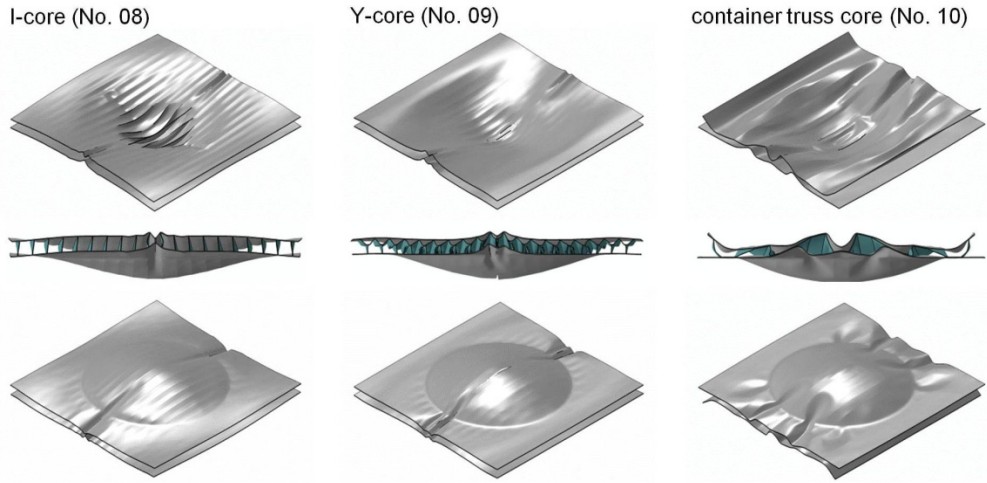

**Figure A3.** Deformation states of the top and bottom sides for prismatic, one-layered, non-auxetic cores No. 08–10 subjected to air-blast loading (t = 0.6 ms).

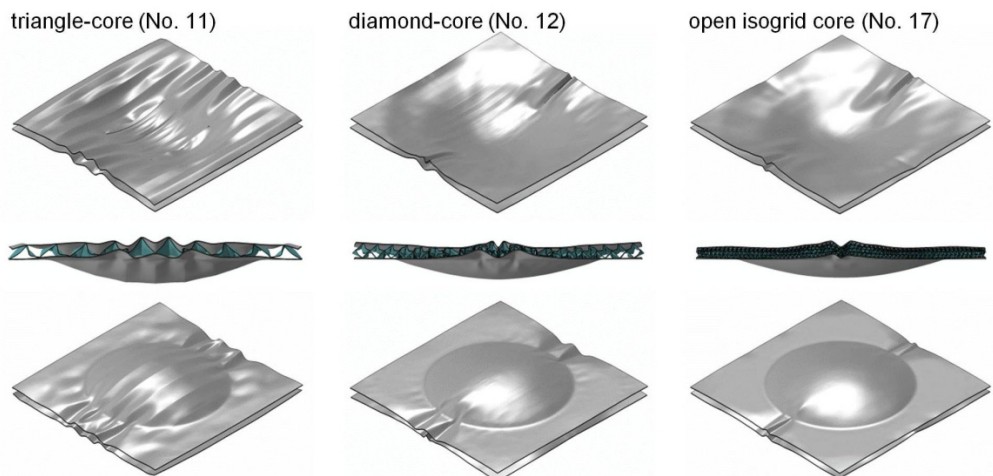

**Figure A4.** Deformation states of the top and bottom sides for prismatic, one-layered, non-auxetic cores No. 11, 12, and 17 subjected to air-blast loading (t = 0.6 ms).

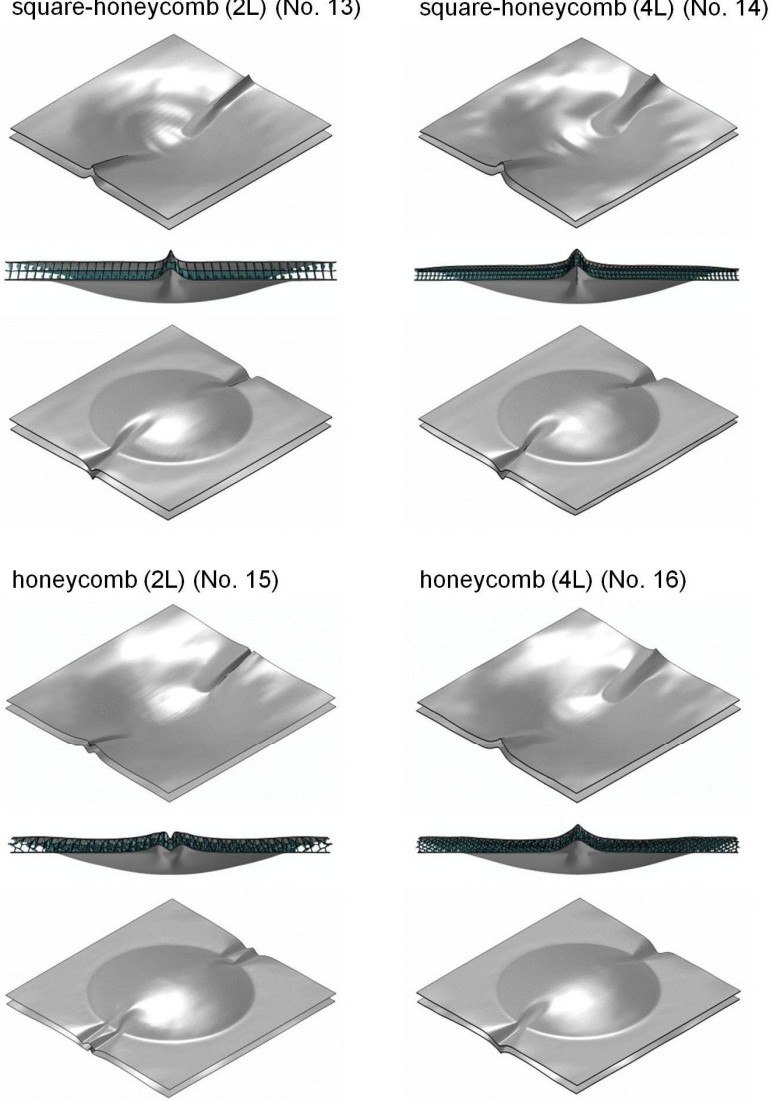

**Figure A5.** Deformation states of the top and bottom sides for prismatic, multi-layered, non-auxetic cores No. 13–16 subjected to air-blast loading (t = 0.6 ms).

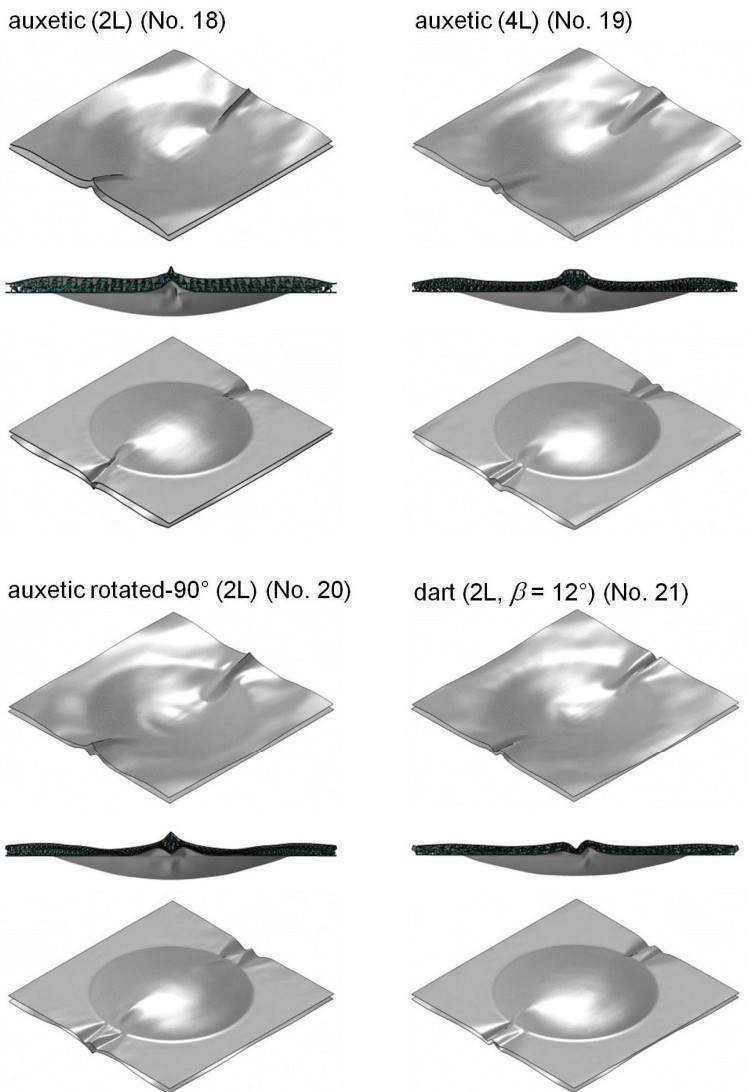

**Figure A6.** Deformation states of the top and bottom sides for prismatic, multi-layered, auxetic cores No. 18–21 subjected to air-blast loading (t = 0.6 ms).

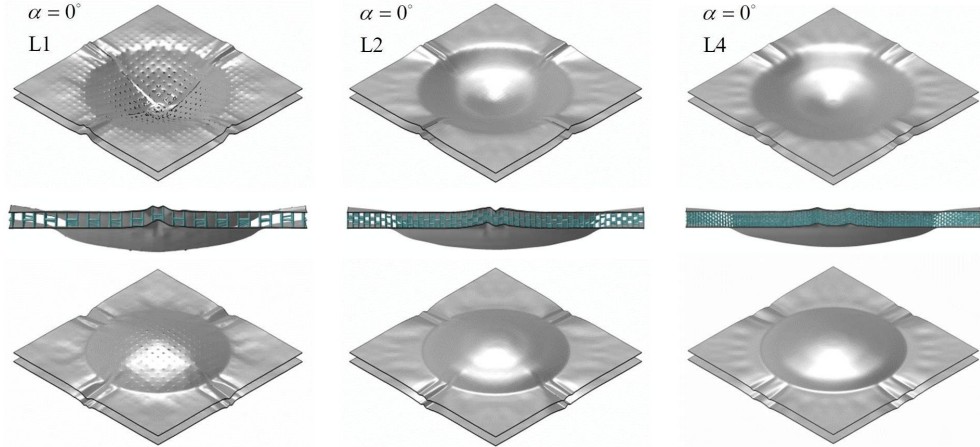

**Figure A7.** Deformation states of the top and bottom sides for 3D lattice, one- and multi-layered, non-auxetic cores with cell angle $\alpha = 0°$ subjected to air-blast loading (t = 0.6 ms).

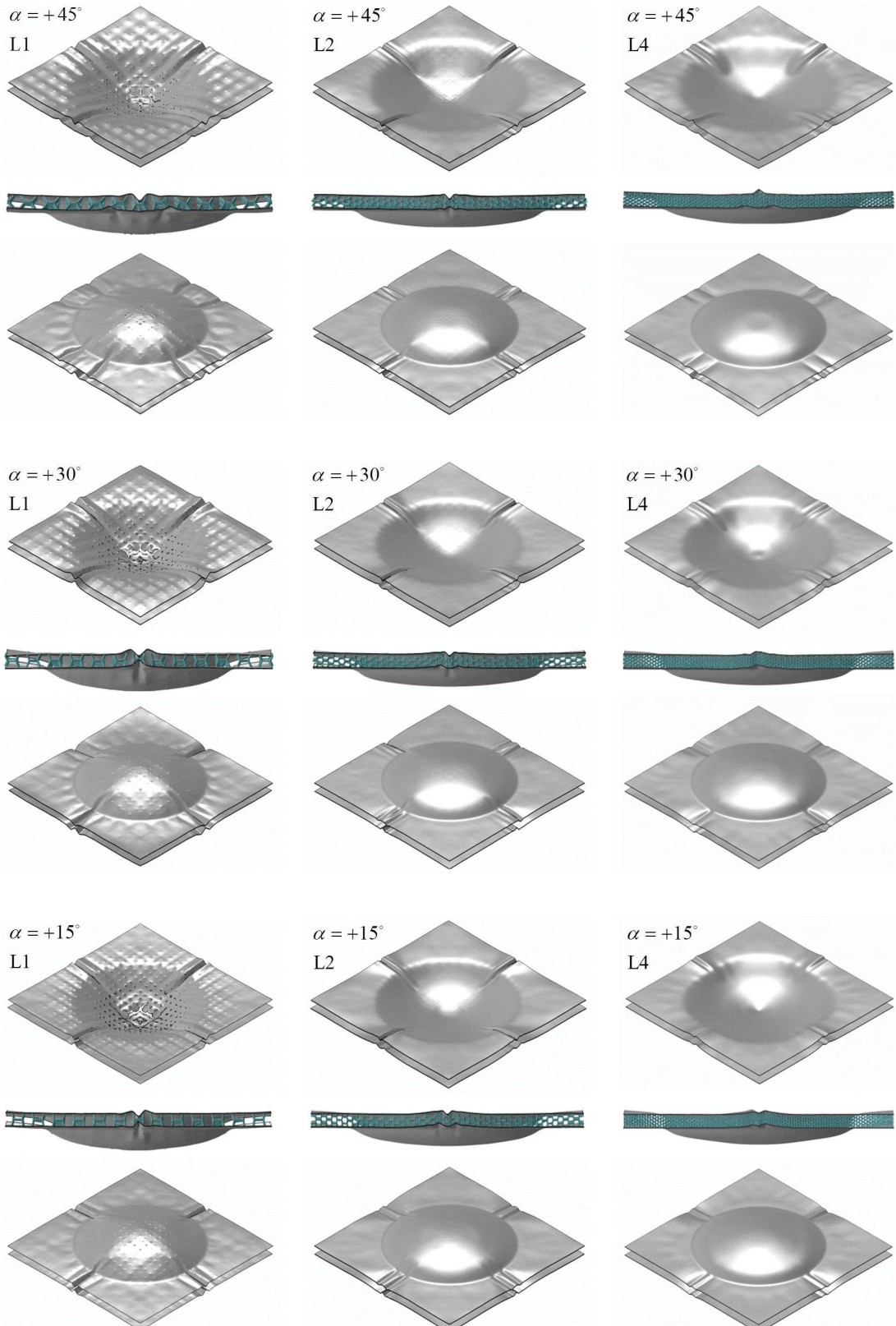

**Figure A8.** Deformation states of the top and bottom sides for 3D lattice, one- and multi-layered, non-auxetic cores with cell angle $\alpha > 0°$ subjected to air-blast loading (t = 0.6 ms).

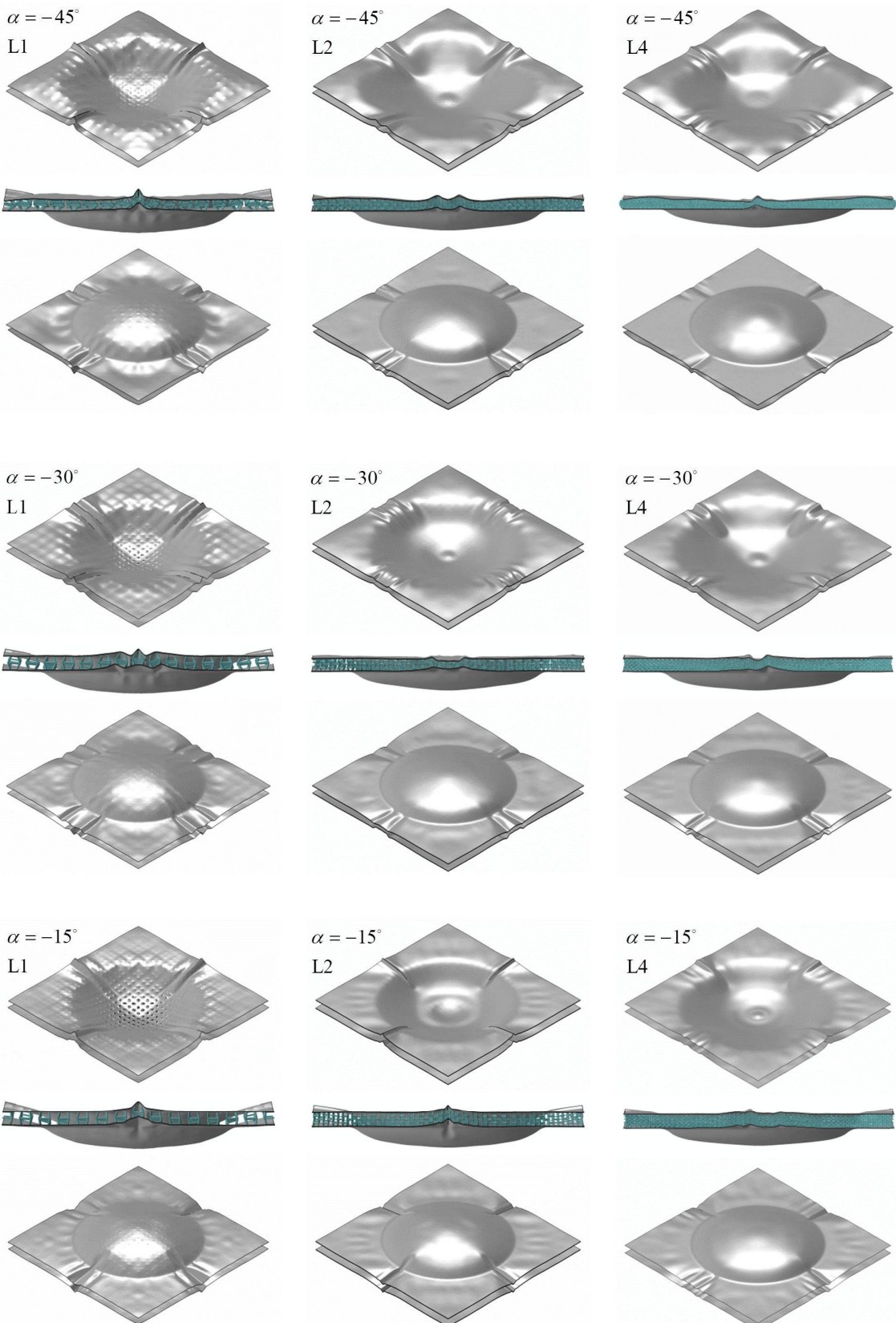

**Figure A9.** Deformation states of the top and bottom sides for 3D lattice, one- and multi-layered, auxetic cores with cell angle $\alpha < 0°$ subjected to air-blast loading (t = 0.6 ms).

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
