# Peer review of "Numerical Investigation of Different Core Topologies in Sandwich-Structured Composites Subjected to Air-Blast Impact"

_applsci, doi:10.3390/app12189012_

Round 1

Reviewer 1 Report

There is little content, which need be revised according to the comment of reviewer in order to meet the requirements of publish. A number of concerns listed as follows:

(1) The abstract should be rewritten to reflect the significance of the proposed work. The current abstract shows a lot of background information.

(2) Please highlight your contributions in introduction.

(3) To explore Comparative results with existing approaches/methods relating to the proposed work.

(4) The authors need to interpret the meanings of the variables.

(5) At line 174, CFRP is what meaning?

(6)  In Table 1. and Table 2.,….. The values of parameters could be a complicated problem itself, how the authors give the values of parameters in the used methods. How to set these parameters values? The authors should provide the reason.

(7) Conclusion: What are the advantages and disadvantages of this study compared to the existing studies in this area?

(8) The inspiration of your work must further be highlighted. Some suggested recent literatures should add. For example,

[1]https://doi.org/10.3390/agriculture12060793

[2]https://doi.org/10.1109/JSTARS.2021.3059451

[3]https://doi.org/10.1016/j.engappai.2022.105139

[4]https://doi.org/10.1007/s10489-022-03719-6

(9) Correct typological mistakes and mathematical errors.

Author Response

General: The authors wish to thank the reviewers for their efforts in making the review and detailed comments. Indeed, these comments have helped us very much to improve the new version of the manuscript. Following the reviewers’ suggestions, the paper has gained a lot. We tried to implement all suggested changes as far as possible.

Reviewer #1

Thank you very much for your very accurate review and your constructive ideas that helped us very much to improve our manuscript.

There is little content, which need be revised according to the comment of reviewer in order to meet the requirements of publish. A number of concerns listed as follows:

(1) The abstract should be rewritten to reflect the significance of the proposed work. The current abstract shows a lot of background information.

Answer: The abstract is rewritten and made more precise.

(2) Please highlight your contributions in introduction.

Answer: In the present version of the manuscript an additional passage in the introduction highlights the novelty aspects of our contribution.

(3) To explore Comparative results with existing approaches/methods relating to the proposed work. (4) The authors need to interpret the meanings of the variables.

Answer: The meaning of the main evaluation parameters are now explained in more detail.

 (5) At line 174, CFRP is what meaning?

Answer: CFRP means carbon fibre reinforced polymer which is defined in line 270 of the submitted manuscript. Following your remark the definition is now put at line 174.

(6)  In Table 1. and Table 2.,….. The values of parameters could be a complicated problem itself, how the authors give the values of parameters in the used methods. How to set these parameters values? The authors should provide the reason.

Answer: The parameters of the conventional core geometries in Table 1 and 2 are based on common values for sandwich structures of this size. (added at the appropriate place in the text). The parameters are set in that way that the total weight, relative core density, core height, outer dimensions, cover thicknesses, etc remain constant to provide a sound comparison of different core topologies.

 (7) Conclusion: What are the advantages and disadvantages of this study compared to the existing studies in this area?

Answer: The advantage is a comprehensive comparative performance overview of representatives of all three principal core types (open and closed prismatic cores as well as lattice cores) in the same loading situation with identical boundary conditions (and both non-/auxetic design).

(8) The inspiration of your work must further be highlighted. Some suggested recent literatures should add. For example,

[1]https://doi.org/10.3390/agriculture12060793

Research on the Time-Dependent Split Delivery Green Vehicle Routing Problem for Fresh Agricultural Products with Multiple Time Windows

[2]https://doi.org/10.1109/JSTARS.2021.3059451

A Hyperspectral Image Classification Method Using Multifeature Vectors and Optimized KELM

[3]https://doi.org/10.1016/j.engappai.2022.105139

Parameter adaptation-based ant colony optimization with dynamic hybrid mechanism

[4]https://doi.org/10.1007/s10489-022-03719-6

Robust visual tracking for UAVs with dynamic feature weight selection

Answer: Thank you very much for your suggestion for additional literature. Unfortunately, none of the papers fits into the scope of our present contribution. Our literature review is very exhausting and already contains 72 references.

 (9) Correct typological mistakes and mathematical errors.

Answer: The authors provided an additional review of the manuscript with regards to typological mistakes and mathematical errors by the research department of our university.

Reviewer 2 Report

1 The abstract should be revised. The significances in engineering field should be highlighted.

2 The authors are suggested to explain the novelty of the paper.

3 For the analysis in Section 3, further step analysis should be added. The authors are suggested to add comments as well as the references below. They are closely related with the present research. Mechanical Systems and Signal Processing, 2023, 182: 109349,  Tribology International, 2021, 164: 107105 .

4 For the conclusions, it is a little long. Several brief points of conclusions are enough. The authors are suggested to rewrite the conclusions.

5 The English should be improved considerately.

Author Response

General: The authors wish to thank the reviewers for their efforts in making the review and detailed comments. Indeed, these comments have helped us very much to improve the new version of the manuscript. Following the reviewers’ suggestions, the paper has gained a lot. We tried to implement all suggested changes as far as possible.

Reviewer #2

1 The abstract should be revised. The significances in engineering field should be highlighted.

Answer: The abstract is rewritten and made more precise.

2 The authors are suggested to explain the novelty of the paper.

Answer: In the present version of the manuscript an additional passage in the introduction highlights the novelty aspects of our contribution.

3 For the analysis in Section 3, further step analysis should be added. The authors are suggested to add comments as well as the references below. They are closely related with the present research. Mechanical Systems and Signal Processing, 2023, 182: 109349,  Tribology International, 2021, 164: 107105 .

Answer: Section 3 focusses on the characterization of the considered core topologies and is already very detailed including an exhausting literature review. The proposed literature stems from the field of lubrication and tribology. Since this contribution deals with the structural mechanics of air-blast-loaded sandwich structures, there is unfortunately no link to our research. For this reason, the authors refrain from citing the proposed additional literature.

4 For the conclusions, it is a little long. Several brief points of conclusions are enough. The authors are suggested to rewrite the conclusions.

Answer: In the discussion section we deliberately created a rather short summary of our simulation results and present potential for future research activities. A significant reformulation or shortening of this section would undercut important aspects of our contribution.

5 The English should be improved considerately.

Answer: The authors provided an additional review of the manuscript by native English-speaking colleagues as well as the research department of our university.

Reviewer 3 Report

This study conducted a numerical analysis of different core topologies in sandwich-structured composites subjected to blast loading. The effects of lattice core topologies exerting auxetic and classical non-auxetic deformation characteristics were investigated in order to illustrate the beneficial properties of auxetic core topologies.

Unfortunately, a conventional FE method was used to analyze the mechanical behavior of the sandwich-structured composites under blast loading. The numerical analysis method adopted is not so innovative, and the description of the FE technique is inadequate. Moreover, the structure of the paper should be improved, and the numerical results in the present work need to be validated by experiments. Some improvements and explanations are needed to improve the quality of the paper.

1. The descriptions in Section 2 and 3 are too long, and some of the contents should be moved to the introduction.

2. The authors sated in Section 5 that “The elastic material parameters and the density of the aluminium alloy are shown in Tab. 3.1”. However, there is no Tab. 3.1 in the paper, but Tab. 5.

3. Figures of the FE models of the sandwich-structured composites should be given in Section 5, including the element meshes, boundary conditions, contacts, etc.

4. Detailed information of the constitutive model of the material should be added, as well as the mechanical properties and constitutive parameters adopted in the material model. Moreover, was the effect of material fracture taken into account regarding the blast impact load?

5. The authors sated in Section 6 that “Apart from the dissipated energy, all quantities reach their absolute maximum within 0.6 milliseconds”. However, as shown in Fig. 17 (a), the peak value on the blue line appeared outside the aforementioned time range.

6. The numerical results in this paper need to be verified by experimental results.

Author Response

General: The authors wish to thank the reviewers for their efforts in making the review and detailed comments. Indeed, these comments have helped us very much to improve the new version of the manuscript. Following the reviewers’ suggestions, the paper has gained a lot. We tried to implement all suggested changes as far as possible.

Reviewer #3

This study conducted a numerical analysis of different core topologies in sandwich-structured composites subjected to blast loading. The effects of lattice core topologies exerting auxetic and classical non-auxetic deformation characteristics were investigated in order to illustrate the beneficial properties of auxetic core topologies.

Unfortunately, a conventional FE method was used to analyze the mechanical behavior of the sandwich-structured composites under blast loading. The numerical analysis method adopted is not so innovative, and the description of the FE technique is inadequate. Moreover, the structure of the paper should be improved, and the numerical results in the present work need to be validated by experiments. Some improvements and explanations are needed to improve the quality of the paper.

Remark: The main objective of the paper was explicitly not to present a novel finite element based numerical approach for the modelling and simulation of blast-loading scenarios. As shown in the paper and our literature review, we employ a commercial finite element code including the state of the art FE routines. The focus of our paper was to provide a comprehensive, FE based and sound comparative performance overview of representatives of all three principal core types (open and closed prismatic cores as well as lattice cores) in the same loading situation with identical boundary conditions (and both non-/auxetic design).

  1. The descriptions in Section 2 and 3 are too long, and some of the contents should be moved to the introduction.

Answer: The state of the art which is presented in Section 2 is extremely important to embed the novelty of the current contribution in the recent literature framework. One page for section 2 is already very concise. The presentation of the different core topologies with their parametrization is very important to readers which are not familiar with such specific sandwich structures. In this way we want to open our research to a wide audience.

  1. The authors sated in Section 5 that “The elastic material parameters and the density of the aluminium alloy are shown in Tab. 3.1”. However, there is no Tab. 3.1 in the paper, but Tab. 5.

Answer: Thank you very much. We corrected this mistake in the present version of the manuscript. 

  1. Figures of the FE models of the sandwich-structured composites should be given in Section 5, including the element meshes, boundary conditions, contacts, etc.

Answer: In order to make more clear how our finite element model is built-up an additional exemplary figure (see fig. 14) is included in the present version of our paper. In this figure we give an impression of the element meshes, boundary conditions, the contact framework, etc. Additionally, we provide more background information on the employed types of elements.

  1. Detailed information of the constitutive model of the material should be added, as well as the mechanical properties and constitutive parameters adopted in the material model. Moreover, was the effect of material fracture taken into account regarding the blast impact load?

Answer: The listing of all material information is beyond the scope of this already very extensive work. The employed material model for the alloy used here is verified by two reference solutions for thin-walled structures. The experimental results of a quasi-static three-point bending test and a dynamic impact test serve as a basis for comparison. All details and results can be found in the reference example “Progressive failure analysis of thin-wall aluminum extrusion under quasi-static and dynamic loads” of the Abaqus online documentation. The basis for the material model is the research of Hooputra et al. at the BMW Group Research and Innovation Center in Munich. Their comprehensive approach takes into account not only the description of the rate-dependent elasto-plastic material behavior but also the influence of various failure mechanisms: ductile failure due to nucleation, growth and coalescence of voids, shear failure due to fracture within shear bands and failure due to necking instabilities. All necessary material data, in particular for characterizing the hardening and strain rate behavior as well as the failure phenomena, are taken from the above-mentioned study. A corresponding short explanation has been added in Section 5.

  1. The authors sated in Section 6 that “Apart from the dissipated energy, all quantities reach their absolute maximum within 0.6 milliseconds”. However, as shown in Fig. 17 (a), the peak value on the blue line appeared outside the aforementioned time range.

Answer: The blue line reaches its maximum just before 0.6 milliseconds. Therefore we have limited all diagrams exactly to this period of observation.

  1. The numerical results in this paper need to be verified by experimental results.

Answer: This contribution deliberately focusses on a numerical study of numerous representative core topologies subjected to air-blast loading. As already mentioned, our goal was not to present a novel finite element based numerical approach for the modelling and simulation of blast-loading scenarios An experimental verification of our numerical results would go far beyond the scope of this already very extensive work. However, the employed numerical models for blast loading as well as modelling of the shock wave are already validated by experiments as shown in our very extensive literature review. However, air blast experiments including TNT are also not trivial and require special facilities. This is the reason, why in this context numerical studies are preferred.

Round 2

Reviewer 1 Report

According to the revised paper, I have appreciated the deep revision of the contents and the present form of this manuscript. There is little content, which need be revised according to the comment of reviewer in order to meet the requirements of publish. A number of concerns listed as follows:

  (1) The main contributions of this paper should be further summarized and clearly demonstrated. This reviewer suggests the authors exactly mention what is new compared with existing approaches and why the proposed approach is needed to be used instead of the existing methods.

(2) More statistical methods are recommended to analyze the experimental results.

(3) How to determine these parameters? The author should give a detailed explanation.

(4) The computation complexity of the proposed method should be clearly described

(5) Although the author has compared many methods, the advantages and disadvantages of the methods, and the incomplete analysis, the author needs to supplement the corresponding experimental analysis.

(6) Please further highlight the introduction, some suggested references should be added to the paper for improving the review’s part.
(7) Please further correct typological mistakes and mathematical errors. There are some grammatical mistakes and typo errors. please proof read from native speaker.

Reviewer 2 Report

accept

Author Response

Thank you very much for your positive review of our corrected manuscript.

Reviewer 3 Report

The paper can be accepted in its present form.

Author Response

(The authors gave the same response as above.)
